# Biomimetic generation of the strongest known biomaterial found in limpet tooth

Robin M. H. Rumney[1], Samuel C. Robson [1,2,3], Alexander P. Kao[4], Eugen Barbu[1], Lukasz Bozycki[1,5], James R. Smith[1], Simon M. Cragg [6], Fay Couceiro[7], Rachna Parwani[4,8], Gianluca Tozzi[4], Michael Stuer [9], Asa H. Barber[4,10], Alex T. Ford [6] & Dariusz C. Górecki [1✉]

The biomaterial with the highest known tensile strength is a unique composite of chitin and goethite ($\alpha$-FeO(OH)) present in teeth from the Common Limpet (*Patella vulgata*). A biomimetic based on limpet tooth, with corresponding high-performance mechanical properties is highly desirable. Here we report on the replication of limpet tooth developmental processes ex vivo, where isolated limpet tissue and cells in culture generate new biomimetic structures. Transcriptomic analysis of each developmental stage of the radula, the organ from which limpet teeth originate, identifies sequential changes in expression of genes related to chitin and iron processing. We quantify iron and chitin metabolic processes in the radula and grow isolated radula cells in vitro. Bioinspired material can be developed with electrospun chitin mineralised by conditioned media from cultured radula cells. Our results inform molecular processes behind the generation of limpet tooth and establish a platform for development of a novel biomimetic with comparable properties.

[1] School of Pharmacy and Biomedical Sciences, University of Portsmouth, St Michael's Building, White Swan Road, Portsmouth PO1 2DT, UK. [2] Centre for Enzyme Innovation, University of Portsmouth, Portsmouth PO1 2DT, UK. [3] School of Biological Sciences, University of Portsmouth, Portsmouth PO1 2DY, UK. [4] Zeiss Global Centre, School of Mechanical and Design Engineering, University of Portsmouth, Portsmouth PO1 3DJ, UK. [5] Laboratory of Biochemistry of Lipids, Nencki Institute of Experimental Biology, Warsaw, Poland. [6] Institute of Marine Sciences, School of Biological Sciences, University of Portsmouth, Portsmouth PO4 9LY, UK. [7] School of Civil Engineering and Surveying, University of Portsmouth, Portland Building, Portland St, Portsmouth PO3 1AH, UK. [8] Carl Zeiss X-ray Microscopy, Pleasanton, CA, USA. [9] EMPA, Swiss Federal Laboratories for Materials Science and Technology, Überlandstrasse 129, 8600 Dübendorf, Switzerland. [10] Present address: School of Engineering, London South Bank University, 103 Borough Road, London SE10AA, UK. ✉email: darek. gorecki@port.ac.uk

nspiration for novel materials is widely found from nature where evolutionarily optimised, biologicals occur. Perhaps the most renowned biomaterial is spider silk, which has a tensile strength of up to 2.9 GPa, outperforming many engineered materials[1]. Owing to such unique characteristics, the development of biomimetic composites via synthetic approaches has been attempted[2–4]. More recently, the teeth of the Common Limpet (*Patella vulgata*) have been demonstrated to have an even higher tensile strength, of up to 4.9 GPa[5]. The mechanical properties of limpet tooth originate from a unique, highly organised composite structure consisting of flexible chitin nanofibers interspaced with reinforcing filamentous crystals of iron oxide in the form of goethite ($\alpha$-FeO(OH))[6–8].

Composite materials often have superior mechanical properties over individual precursor components[9]. Fully synthetic composites with advantageous combinations of mechanical properties are widely used (e.g. Kevlar) but the manufacturing processes can be toxic and the materials themselves difficult to recycle[10–13]. In contrast, biologically derived materials have the benefit of offering inherent sustainability. The disadvantages of synthetic composites necessitate new materials with superior mechanical performance that meet the key modern engineering challenges of affordability, durability and sustainability through re-use and recycling[14]. This set of challenges may be answered with biomimetics. Indeed, a bioinspired material based on chiton teeth has recently been developed[15].

The composition and structure of limpet tooth material have been studied for over a century. Specifically, the presence of a chitin scaffold was established in 1907[16]; goethite as a key constituent was identified in the 1960's[6] and directionally arranged nanofibrous crystals of goethite within a highly organised chitin matrix composing each tooth were described in 1980's[7]. However, it is only recently that the processes which underlie limpet tooth formation and are critically important for the development of biomimetics, could be unravelled using molecular approaches.

RNA-seq has shown that the radula transcriptome from the freshwater snail *Tylomelania sarasinorum* contains groups of genes associated with vesicular secretion, chitin binding and iron transport[17]. Proteomics on radulae from the limpet *Cellana toreuma* revealed the presence of ferritins in the teeth, while GTPases were identified as the predominant goethite binding proteins[18]. A detailed study on the chiton species *Cryptochiton stelleri* characterised deposition of ferrihydrite in the tooth cusp, which transforms to magnetite between just a few rows of teeth[15]. As the radula is structured as a conveyor belt for development, microdissection combined with molecular analyses can reveal the step-wise formation of limpet teeth, including chitin scaffold formation and gradual inclusion of goethite crystals[7].

In this study, we decipher the molecular mechanisms behind tooth formation in the Common Limpet *Patella vulgata*, recreate this process in organotypic cultures and exploit its fundamental principles to create an analogous biocomposite of interlaced chitin fibres and iron oxide in the form of haematite ($Fe_2O_3$) with significant strength, using an acellular system. This technology is a breakthrough towards exploiting biological fabrication processes to manufacture the strongest known biomaterial.

## Results

**De novo organogenesis of limpet radula and cell organisation in vitro**. The limpet radula is organised as a conveyor belt for tooth development, which begins at the blind end of the formation zone (FZ), which is a bulb-shaped soft tissue structure (Fig. 1a), as recently described in a related patellid[19]. The FZ must continuously generate teeth throughout the limpet lifespan, and it has been shown in other gastropods that teeth emerge from this region in their final form[20]. In the limpet, the four distinct stages of teeth[21] correspond to immature (Stage I, first 15–20 rows without any signs of mineralisation), early maturing (Stage II, the next 12 rows), late maturing (Stage III, the next 30 rows) and mature (Stage IV, the remaining rows) (Fig. 1b–e). These stages differ in the extent of tooth development and mineralisation and can be readily identified by light microscopy (Fig. 1a–e, Supplementary Movie 1), with the changes in mineral density along the length of the organ visualised by high-resolution X-ray computed tomography (XCT) (Supplementary Movie 2).

Whereas other studies have characterised the structure and composition of limpet tooth ex vivo[7,15,21,22], our aim was to create limpet tooth material under laboratory conditions. To this end, we developed novel cell culture protocols to grow radula tissue and primary cells. When the isolated FZ is maintained in vitro, it starts producing a new ribbon of Stage I radula (Fig. 1f–h). It emerges from the FZ after 5 to 7 days in culture, and the new radula continues to grow so that at two weeks there are up to 12 rows of teeth. In the presence of Fe(II)SO$_4$ in the medium teeth undergo spontaneous mineralisation (Fig. 1i, Supplementary Movie. 3).

We established methods allowing cell populations from each stage of the limpet radula to be maintained in vitro. Cells spontaneously migrated from Stage II radula, but survived only a few days (Fig. 1j). In contrast, cells obtained by enzymatic dissociation could be maintained for longer. Coating tissue culture glass with limpet haemolymph increased the cell viability (Fig. 1k). Cells from the FZ and Stages I-IV could be maintained for over two weeks on haemolymph-coated glass (Fig. 1l–p). Remarkably, a suspension of cells isolated from the whole radula (FZ to Stage IV) grown on haemolymph-coated glass, assembled, after two weeks, into structures with notable 'head' and 'tail' regions (Fig. 1q–s), in a process akin to neo-organogenesis. Furthermore, cultures of cells ($n = 3$ independent replicates) isolated from the FZ and maintained for 6 weeks in culture spontaneously generate individual limpet teeth. These teeth emerged from what looked like small clusters or even individual cells attached to the culture dish, were resembling those formed in the radula but significantly smaller ($\sim$20 $\mu$m), yet morphologically similar in size and shape (Fig. 1t).

**Differential gene expression reveals functional specialisation across the radula**. Total RNA was extracted from the FZ and each of the subsequent four radula stages. RNA isolated from the foot muscle was used as a control, with all 6 tissues extracted from 5 individual limpets. RNA sequencing generated 871,497,501 paired-end reads across the samples with average Q20 scores of 95.6% (Supplementary Table 1). In total, 866,226,142 quality trimmed paired-end reads were used for de novo transcriptome assembly. The final assembly consisted of 464,975 transcripts, clustered into 272,735 "genes" by the Butterfly portion of the Trinity algorithm. Following filtering, the final analysis was conducted based on 36,806 transcripts (28,420 genes) (Supplementary Fig. 1a, Supplementary Data 1).

Clustering of samples based on expression levels across all transcripts was clearly demonstrated within tissue groups (Supplementary Fig. 1b). Principal component analysis of variation in gene expression across the entire dataset identified clear differences between the tissues, with the FZ and radula regions accounting for the majority of variance (PC1, 26.39%), while dissimilarity between the muscle and the FZ accounted for the second-largest degree of variance (PC2, 23.38%) (Fig. 2a). Delineated sequential changes in expression were clearly observed from Stage I to Stage IV, potentially indicating successive gene expression changes throughout radula development (Fig. 2a, b).

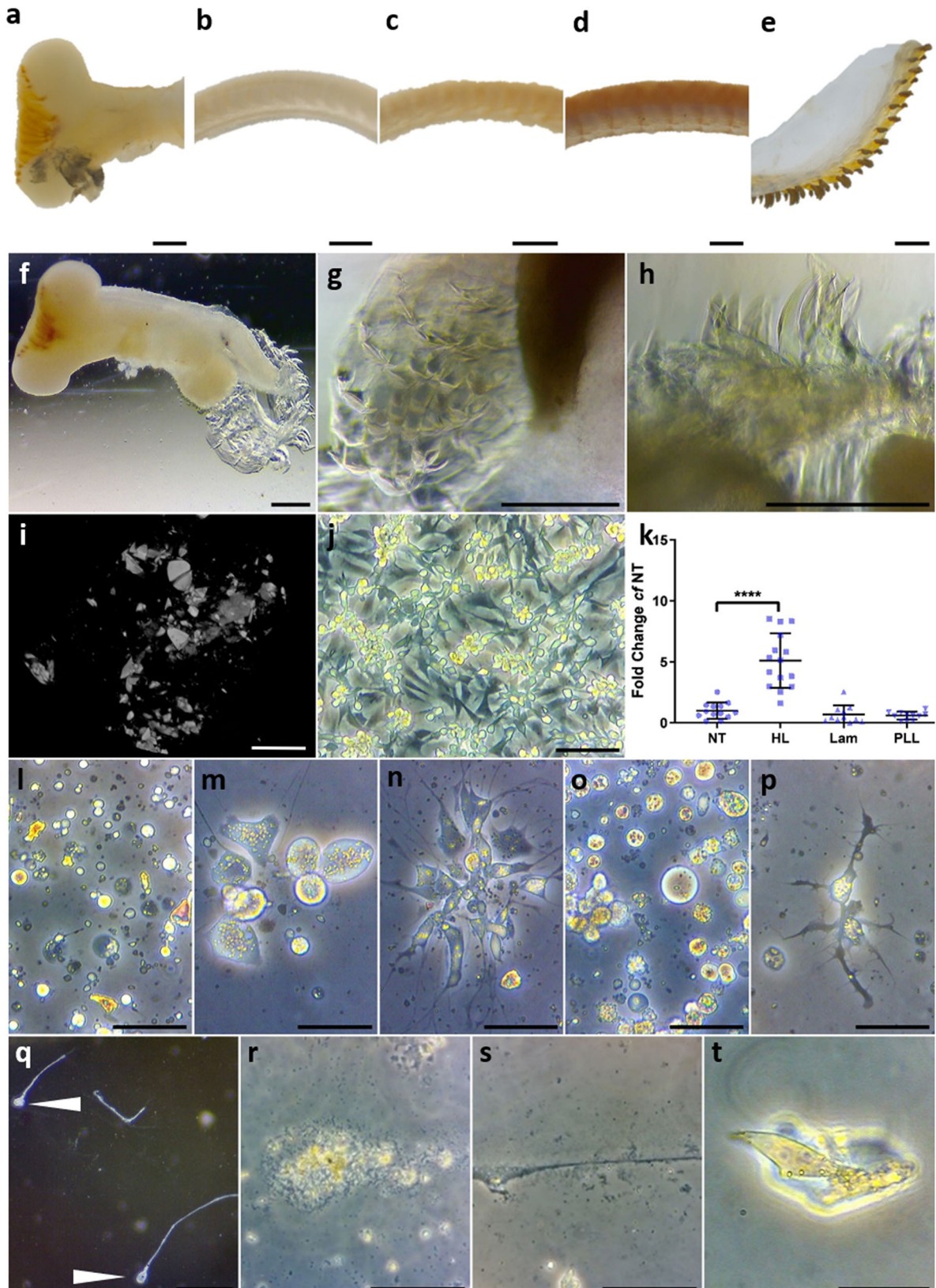

When comparing transcript expression across the muscle, the FZ and radula (all stages combined), 6949 transcripts were expressed in all three regions. Unique gene expression patterns were seen, with the muscle (4,348 transcripts), the FZ (1331 transcripts) and radula (3467 transcripts) (Fig. 2c). Differential gene expression analysis was conducted to identify genes significantly up- or downregulated in the FZ and radula sections compared to the foot muscle (Supplementary Data 2 and 3). Comparison between genes differentially expressed at each radula stage identified clear overlap of genes common across the radula (1,078 downregulated and 301 upregulated), but also genes specifically expressed in one specific stage only, with the FZ

**Fig. 1 The Formation Zone is critical in radula development and regeneration.** Light microscopy images showing **a**, the Formation Zone (FZ) and **b**–**e**, Stages I to IV of the limpet radula respectively (scale bar = 500 μm). **f**–**h** Images of the isolated FZ in vitro, including higher magnification of regenerated Stage I radula ($n = 36$ FZ from 6 independent cultures, scale bars = 500 μm). **i** Reconstructed X-ray computed tomograms depicting mineral deposition within Stage I radula regenerated in the presence of Fe(II)SO$_4$ ($n = 12$ FZ were treated with Fe(II)SO$_4$ with one selected for XRM, scale bar = 100 μm). **j** light microscopy image of cells migrated directly from Stage II radula ($n = 3$ donor limpets, scale bar = 50 μm). **k** FZ cell viability on differently coated glass surfaces (NT = No Treatment; HL = Hemolymph; Lam = Laminin; PLL = Poly-l-lysine) ($n = 16$ wells per group across 4 independent donor limpets, $p = 7.0858$ E-10, Univariate analysis with Tukey's post-tests *cf* NT, data are presented with individual data points and mean with standard deviation error bars). Light microscopy images of cells isolated from **l**, the FZ and **m**–**p**, Stages I to IV of the radula respectively on haemolymph coated glass after two weeks of culture in vitro (scale bar = 50 μm). Light microscopy images of self-organised structures generated by cells isolated from the FZ and all four radula Stages I to IV pooled together, **q** visualised at macro scale with arrows denoting the 'head' ends (scale bar = 500 μm) and **r**, **s**. at microscale showing 'head' and 'tail' respectively ($n = 3$ donor limpets, scale bar = 20 μm). **t**, Individual limpet tooth derived from primary FZ cells passaged after 6 weeks in culture ($n = 3$ donor limpets scale bar = 20 μm).

showing the greatest gene expression activity compared with the muscle (Fig. 2d, e). Expression patterns of genes relevant for tooth formation, e.g., with putative roles in chitin and iron metabolism, were identified and their expression profiles confirmed by qPCR (Supplementary Data 3 and Supplementary Table 2, Fig. 2f, g; Supplementary Fig. 1c, d).

Transcripts associated with chitin synthases (*CHS*) were found along the radula, from the FZ to Stage IV, and four genes were identified as having differential expression patterns. The FZ highly expressed one transcript mapping to both *CHS1* and *CHS8* sequences (DN204240) and another which mapped to *CHS1* and *CHS4* (DN207345). Stage I expressed another *CHS1* transcript (DN209426) that was absent from the FZ and was the highest expressed chitin synthase of any stage in the radula. Expression of *CHS* was generally lower along the rest of the radula, although the DN209426 variant of *CHS1* remained elevated in Stage II, DN207345 for *CHS1/4* peaked in Stage III and low expression of DN204240 corresponding to *CHS1/8* was also present in Stage IV (Fig. 2f, Supplementary Fig. 1c).

Three chitinase 1 (*CHIT1*) transcripts, were identified along the length of the limpet radula. Two transcripts, DN204538 and DN209778 were highly expressed in the FZ with sequential expression decreases along the radula stages. The third *CHIT1* transcript (DN209778) was expressed at high levels all along the radula, except for Stage I (Fig. 2f, Supplementary Fig. 1c).

The immature non-mineralised Stage I radula expresses a melanotransferrin (*TRFM*) transcript. Stages I and II express a transcript for heavy metal binding protein HIP. Furthermore, ascending gradients of expression, from the FZ to Stage IV, are found, for the divalent metal transporter Cyclin and CBS domain divalent metal cation transport mediator 2 (*CNNM2*), for Ferric-chelate reductase (DN211629 and DN211756) and for the metalloreductase six-transmembrane epithelial antigen of prostate 3 (*STEAP3*). A complementary descending gradient of gene expression occurs from the FZ to Stage IV for Hephaestin-like 1 (*HEPHL1*), (Fig. 2g, Supplementary Fig. 1d). Taken together, the differential gene expression profiles along the radula length from the FZ to Stage IV show a consistent pattern, where reductase expression is increased and oxidase expression is decreased.

Moreover, previous studies have identified genes with higher expression in the radula than in muscle or shell mantle[17,18,23]. Differential expression of such candidate genes across radula stages was analysed in depth (Supplementary Fig. 1e). Expression of the morphogenic patterning gene Aristaless Related Homeobox (*ARX*) and the chitin-binding protein Di-N-acetylchitobiase (*DIAC*) were specifically enriched in the FZ only. An additional morphogenic patterning gene, Gastrulation Brain Homeobox 2 (*GBX2*) was found from Stage I to Stage IV radula, together with genes encoding the iron storage protein Ferritin (*FRIS*), the enzyme Superoxide dismutase [Cu-Zn] (*SODC*) and the matricellular Secreted protein, acidic and rich in cysteine (*SPARC*). The

transcription factor Erythroblast Transformation Specific 1 (*ETS1*) transcription appeared up-regulated at Stage III. There are two subsets of transcript encoding chitin-binding PIF proteins; one subset (*PIF_1*) is more highly expressed in Stage IV radula than elsewhere, while another (*PIF_2*) is predominantly expressed in muscle. *HES1* gene expression was also markedly upregulated at Stage IV. Other genes, including those encoding chitin-binding proteins (other than *PIF*) were not differentially expressed across the radula.

**Replication of in vivo mechanisms of limpet tooth generation by cell populations in vitro.** Limpet teeth are initially formed as chitin scaffolds that acquire iron oxide mineral through each developmental stage of the radula[8,19,21]. X-ray computed tomography (XCT) reveals the pattern of mineral acquisition between Stages II and IV with the accumulation of X-ray dense mineral surrounding tooth scaffolds at Stage II, mineralisation within the posterior edge at Stage III and the anterior edge at Stage IV (Fig. 3a–c). Importantly, under conditions established by us, Stage III and IV radula cells maintained for 14 days in vitro, deposited dense extracellular stellate crystals visible under light microscopy following Prussian blue staining (Fig. 3d). Analysis of control goethite and samples from in vitro cultures by transmission electron microscopy with energy dispersive X-ray microanalysis (TEM-EDX) revealed the co-localisation of iron and oxygen (Fig. 3e–j), consistent with the presence of goethite in limpet teeth.

To better characterise these processes, chitin, chitinase activity and iron in lysates collected from each radula stage ex vivo and from cell populations maintained in vitro were measured. All measurements were standardised to protein levels. Stage II radula was found to contain the highest level of chitin (Fig. 3k), which is consistent with the high expression of chitin synthases (Fig. 2f). Chitin synthase activity was quantified in cell lysates from one-day-old cultures following incubation with the chitin precursor N-acetylglucosamine[24] and lysates collected from Stage II cells contained significantly more chitin (Fig. 3l). This indicates the presence of active chitin synthase in Stage II radula cells in vitro, consistent with the pattern of chitin synthase gene expression (Fig. 2f, Supplementary Fig. 1c) and with Stage II containing the most chitin in vivo (Fig. 3k).

Chitinase activity was quantified using substrates specific to different enzymes: exochitinases (Supplementary Fig. 2a–c), chitobiosidases that hydrolyse chitobiosides (dimers of β-1,4-linked glucosamine units) (Supplementary Fig. 2d–f), and endochitinases (Supplementary Fig. 2g–i). Measurements were taken from extracts of radula stages ex vivo and lysates collected from cells after 1 and 14 days in culture. Exochitinase activity was found to be higher in Stage II radula than the FZ (Supplementary Fig. 2a, $p = 0.0434$). In day 1 lysates, exochitinase activity was higher in cells from the FZ and Stages I-II than in Stages III-IV

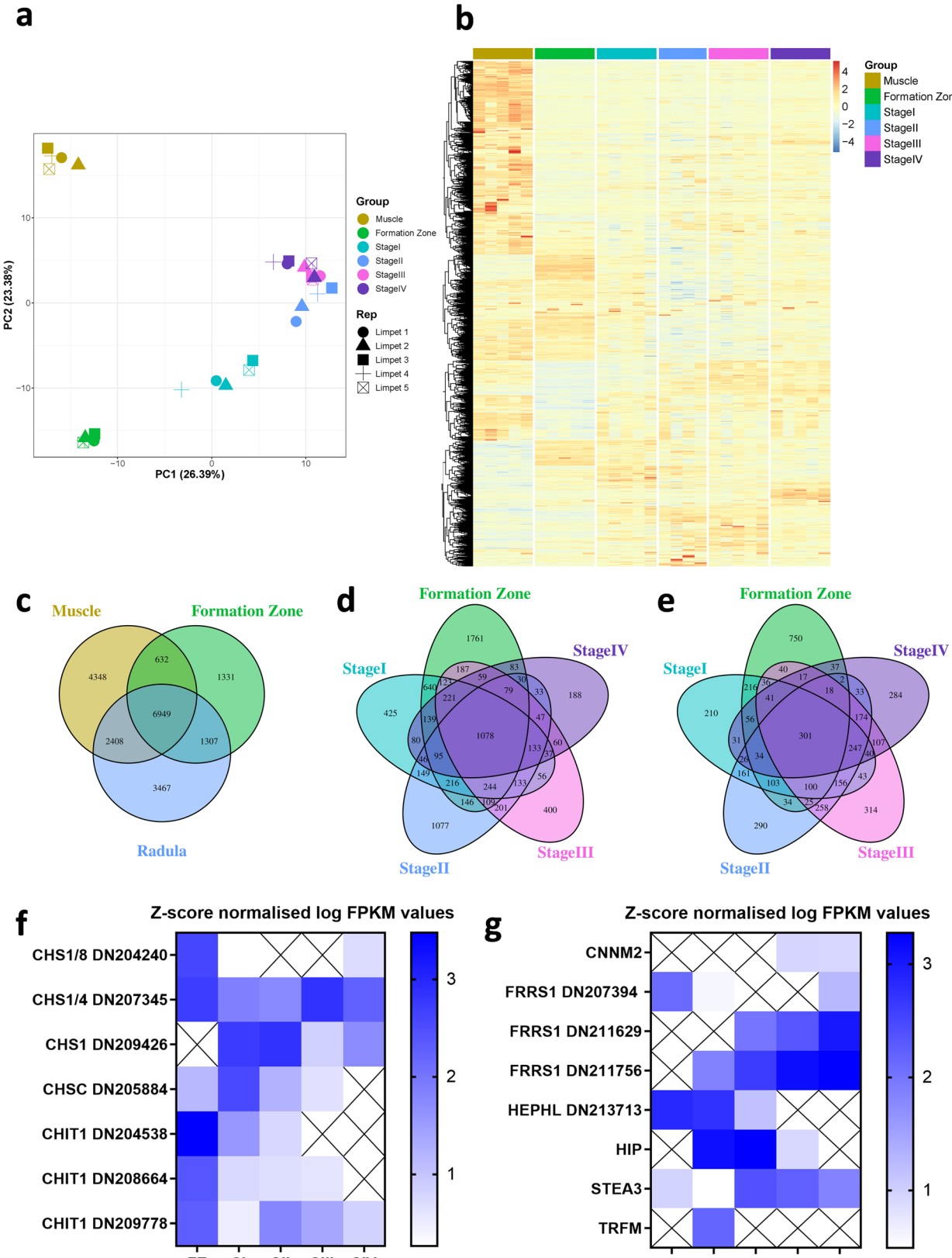

**Fig. 2 The radula transcriptome and expression of key genes controlling limpet tooth formation. a** Principal component analysis (PCA) demonstrating the relatedness of transcript expression between limpet muscle, the FZ and radula Stages I to IV. **b** Heat map showing the abundance (Z-score normalised log FPKM for RNA seq) for all genes with differential expression across the limpet muscle, the FZ and radula transcriptomes. **c** Venn diagram showing overlap of gene expression in the FZ, the radula and muscle. **d–e** Overlap between the genes upregulated, and downregulated respectively, in the radula compared to muscle, which was used as a common control. **f–g** Log FPKM values for RNA seq for genes involved in chitin and iron metabolism, respectively. CHS Chitin synthase, CHIT Chitinase, CNNM2 Cyclin and CBS domain divalent metal cation transport mediator 2, FRRS1 Ferric chelate reductase 1, HEPHL Hephaestin-like protein 1, HIP Heavy metal-binding protein HIP, STEA3 metalloreductase six-transmembrane epithelial antigen of prostate 3, TRFM Melanotransferrin.

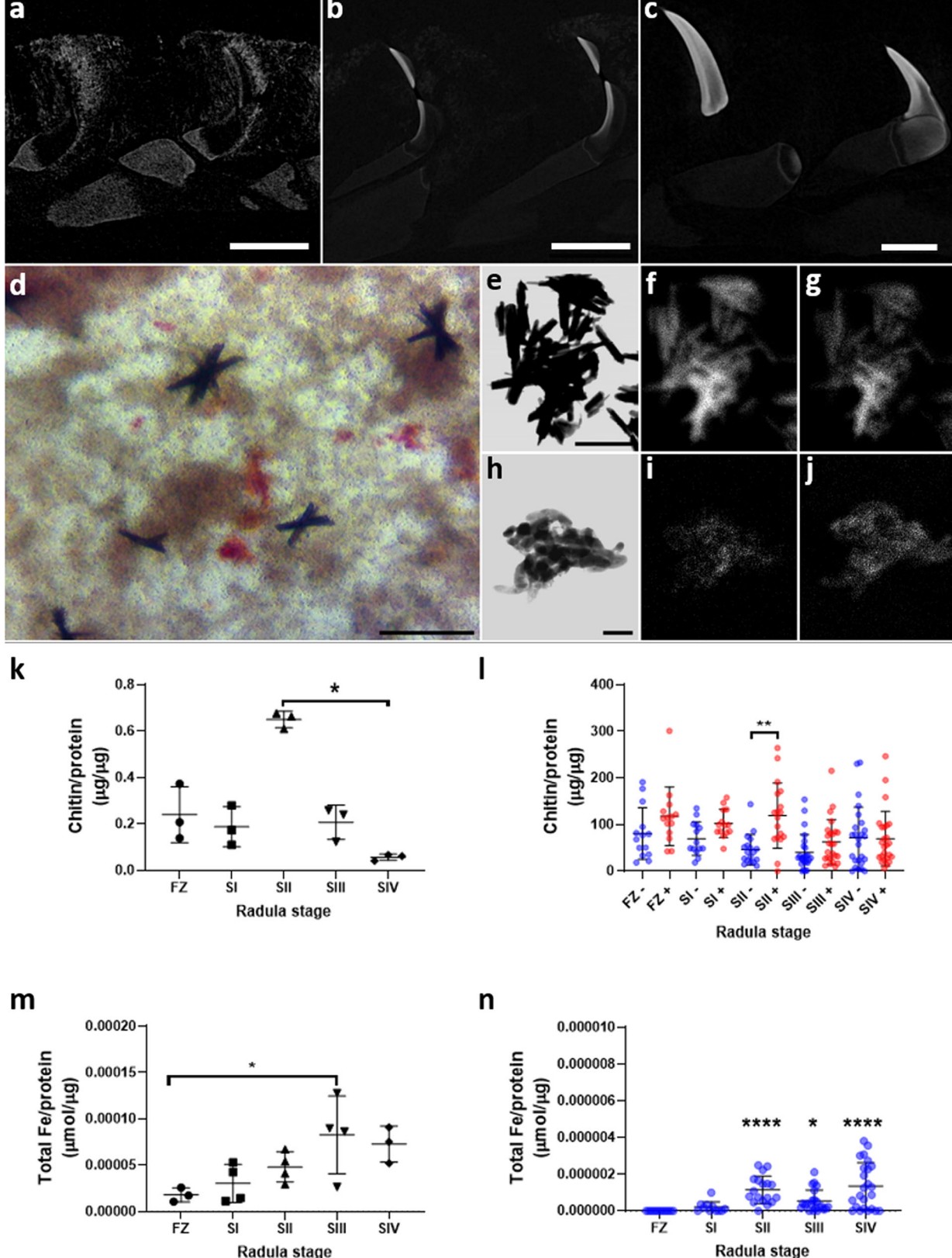

(Supplementary Fig. 2b, $p = 0.002$). In lysates from 14-day cell cultures, exochitinase activity was no longer significantly higher in Stage II populations, but remained elevated in populations from the FZ and Stage I cultures compared to Stages III-IV (Supplementary Fig. 2c, $p = 0.0018$). Chitobiosidase activity was higher in Stages III and IV than the FZ (Supplementary Fig. 2d,

$p = 0.0002$ and $p = 0.00161$) and higher in Stage III than in Stages I and II (Supplementary Fig. 2d, $p = 0.0006$ and $p = 0.0029$). Chitobiosidase activity was higher in lysates from day 1 cells at Stage I than Stage III (Supplementary Fig. 2e, $P = 0.0309$); by day 14 in culture, activity had abated with no differences between cell populations (Supplementary Fig. 2f). Endochitinase activity was

**Fig. 3 The stages of radula development in vivo are recapitulated structurally and molecularly by cells in vitro. a–c** Reconstructed images from X-ray computed tomography scans of teeth from radula Stages II to IV respectively (scale bar = 200 μm). **d** Light microscopy image of stellate crystal formations in the extracellular matrix surrounding Stage III and Stage IV radula cells after two weeks culture in vitro (n = 3 donor limpets, scale bar = 20 μm). **e** TEM-EDX microscopy on control goethite crystals reveal the co-localisation of iron (**f**), and oxygen (**g**) (n = 2 donor limpets, scale bar = 1 μm). **h,** TEM-EDX microscopy on crystals isolated from limpet cell cultures reveal the co-localisation of iron (**i**), and oxygen (**j**). **k** Chitin quantities in each radula stage ex vivo and normalised to protein (* p = 0.0195, Dunn's multiple comparison test, n = 3 independent biological replicates with one measurement per limpet). **l** Chitin quantities in lysates taken from cells after 24 h in vitro and incubated with either vehicle (blue) or with the chitin precursor NaGluc (red) (***p = 0.000365, Univariate analysis with Tukey's post hoc test). **m** Iron quantities in each radula stage ex vivo and normalised to protein (* = p = 0.0425, Dunn's multiple comparison test). **n** Iron quantities in lysates taken from cells after 14 days in vitro (****Stage II v the FZ p = 0.000002; *Stage III v the FZ p = 0.011; ****Stage IV v the FZ p = 3.3759E-8; Univariate analysis with Dunnett's post-hoc test). Data shown with mean and standard deviation respectively over 3 biological replicates.

highest in Stage III (Supplementary Fig. 2g, $p = 0.0018$), however endochitinase activity was absent from cell lysates at days 1 and 14 (Supplementary Fig. 2h, i). These data demonstrate continued exochitinase and chitobiosidase activity in isolated limpet radula cells in vitro reflecting chitinase activity in vivo.

These data suggest that most chitin is produced in Stage II and not the FZ, while most chitinase activity occurs in Stages II-III. The lack of chitinase activity in the FZ, concomitant with high expression of different chitinase genes there may suggest a lack of translation, or the generation of protein precursors requiring activation.

Iron (including total Fe, $Fe^{2+}$ and $Fe^{3+}$) quantified using a colorimetric assay was detected in lysates from all radula sections. Total iron/protein was higher in Stage III radula ex vivo than in the FZ (Fig. 3m, $p < 0.05$), while in day 14 cells in vitro total iron was higher in Stages II to IV than the FZ (Fig. 3n, $p < 0.05$ the FZ $cf$ Stage III, $P < 0.0001$ the FZ $cf$ Stages II and IV). $Fe^{2+}$ was highest in Stage II (Supplementary Fig. 3a), while $Fe^{3+}$ was highest in Stages III and IV radula ex vivo (Supplementary Fig. 3b).

These data are consistent with differential gene expression that show changes in ferroxidase and reductase expressions that reflect goethite deposition in more mature radula stages. Overall, these results demonstrated that cells isolated from the radula maintain the activities required for the deposition of limpet tooth material in vitro.

**Generation of synthetic limpet tooth material.** Understanding the underlying molecular processes of limpet tooth formation in vitro facilitated attempts to produce biomimetic material with similar mechanical properties. While previous studies have used radioactive tracers[25] or taken solely histological, structural-mechanical approaches[7,15,21,26,27], we have analysed transcriptomics and in vitro biogenesis to inform our approach. We took inspiration from other forms of biomineralization where a scaffold is set by specialised cells and then mineralised by secreted factors, as exemplified by bone where osteoblasts deposit a collagen matrix, which is then mineralised by secreted matrix vesicles containing the molecular apparatus required for mineralisation[28]. Furthermore, analysis of orthologs of genes linked to vesicular transport and release identified a differential pattern of their expression across the radula sections (Supplementary Fig 4), supporting a putative vesicular mechanism.

To investigate whether radula cells can mineralise chitin scaffolds in an analogous process, we collected and filter-sterilised conditioned media (CM) from 7-day-old cultures of Stage III and IV radula cells, which had been maintained in basal media with no additional iron. Chitin from shrimp shells was solubilised in 1,1,1,3,3,3–Hexafluoroisopropanol (HFIP) and used for electrospinning[29]. Incubation of electrospun chitin scaffolds with the CM for 14 days engendered extensive iron deposition visible across whole chitin discs (Fig. 4a, b). In contrast to other attempts at making chitin based composite materials, which substitute chitosan in place of chitin[15], our biomimetic material is more similar in composition to the actual limpet tooth. SEM visualised the presence of crystals within these CM-treated chitin scaffolds (Fig. 4c, supplementary Fig. 5). Energy-dispersive X-ray spectroscopy demonstrated that the crystals were composed of iron oxide (Fig. 4 d–g). The precise form of deposited iron oxide was determined by Raman spectroscopy, whereupon the Raman spectra were consistent with hematite ($Fe_2O_3$) (Fig. 4h, i).

These data demonstrate that extracellular components released by radula cells play a key role in the chitin mineralisation and, importantly, these processes are both reproducible and scalable in vitro by using cell-conditioned medium. The stiffness of such mineralised chitin scaffolds was quantified using atomic force microscopy (AFM). The results reveal that threads within the meshwork of chitin scaffold mineralised using CM from Stage III-IV radula cells had force vs. distance curves comparable to solid sheets of polymethylpentene (PMP) and polypropylene (PP), used as reference materials (Fig. 4j). At 25.6 GPa, the reduced modulus of CM-treated chitin is much greater than non-treated chitin at 0.034 GPa ($P < 0.05$), but within the range of PMP at 53.8 GPa (Fig. 4k). These data indicate over 1500-fold increase in the tensile stiffness of electrospun chitin following mineralisation and confirm the critical role of secreted factors in the radula mineralisation process. By promoting the deposition of iron oxide and changing the mechanical properties of chitin scaffolds, our approach led to creation of a novel biomimetic composite material.

## Discussion

We have characterised the molecular mechanisms behind tooth formation in the Common Limpet (*Patella vulgata*), recreated this entire development ex vivo and in vitro and exploited the established principles to create a high strength biocomposite of interlaced chitin fibres and iron oxide crystals in an entirely acellular process. This technology is a breakthrough towards exploiting biological fabrication processes to manufacture this strongest known biomaterial.

To create limpet tooth composite under laboratory conditions we developed novel cell culture protocols allowing us to grow radula tissue and primary cells. Isolated FZ maintained in vitro retained its in vivo properties, producing a ribbon of Stage I radula with teeth undergoing spontaneous mineralisation in the presence of Fe in the medium. While a few teeth could be simply brought out by migration of the subradular epithelium, the observed generation of a ribbon with 12 rows of teeth makes this explanation unlikely. This radula generation ex vivo is a novel finding, which confirms the unique ability of gastropod tissues to regenerate in culture[30].

Furthermore, cultures of individual cells isolated from the FZ and maintained for 6 weeks in culture spontaneously generated individual limpet teeth. This ability to create an entire limpet tooth in culture, in the absence of the complex multicellular organ

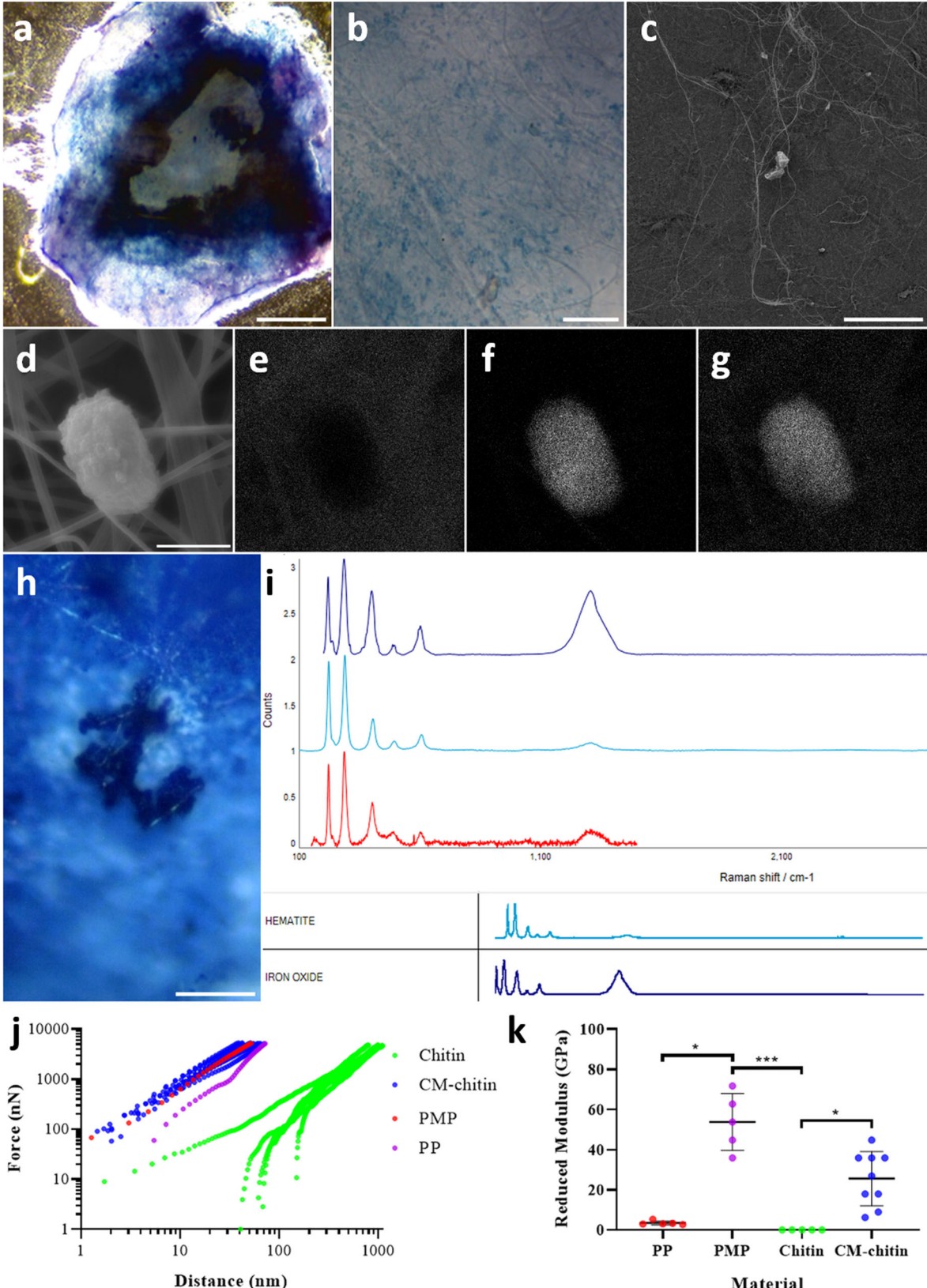

such as the FZ, demonstrated a remarkable stem cell potential. Of the known types of cells in the FZ, membranoblasts generate the radular membrane while odontoblasts are highly specialised cells that individually or in groups form one tooth[19,31]. It is possible that odontoblasts are long lived and were responsible for tooth formation in vitro[32]. Another possibility is that undifferentiated,

highly mitotic cells found in the lateral part of the FZ, presumed to be replenishing the odontoblast and membranoblast populations, are the source of multipotent cells responsible for tooth formation in vitro[25].

Equally remarkably, a suspension of cells isolated from the whole radula, after two weeks assembled into structures

**Fig. 4 A new material based upon limpet tooth generated in vitro. a, b** Electrospun chitin scaffolds treated with cell-free conditioned media from Stage III and IV radula cell cultures and stained with Prussian blue to label iron containing deposits (scale bars = 1 mm and 100 µm respectively). **c** SEM on chitin scaffolds treated with Stage III/IV radula cell conditioned medium showing distribution of crystals (scale bar = 50 µm). **d** High magnification EDX showing **e** an absence of carbon, **f** the presence of iron and **g** the presence of oxygen (scale bar = 1 µm). **h** Site of Ramen spectroscopy on mineralised chitin scaffolds (scale bar = 20 µm). **i** Ramen spectra showing presence of iron oxide most likely in the form of hematite. **j** AFM data showing that electrospun chitin treated with Stage III/IV radula cell conditioned medium (CM-chitin) has a stiffness comparable to sheets of polypropylene (PP), polymethylpentene (PMP) and glass. **k** Reduced modulus of PP, PMP, chitin and CM-chitin calculated from Force (nN) and Distance (nm) data with mean and standard deviation ($n = 5$ force moduli from 5 force vs. indentation$^{3/2}$ plots for PP, PMP and chitin and 10 from CM-chitin with CM from two independent donor limpets; *PP v PMP, $p = 0.016$; ***PMP v chitin $p = 0.0002$; *chitin v CM-chitin $p = 0.0103$; Kruskal-Wallis test).

containing a distinct bulbous "head" and an elongated "tail". It is currently not known whether these structures formed in a process involving migration and aggregation of the isolated cells or rather proliferation of specific cells, such as those aforementioned undifferentiated mitotic FZ cells, which should have capability to differentiate quickly, and their subsequent precise arrangement in response to some specific cues.

Analysis of variation in gene expression identified clear differences between the muscle and radula, with further diversity across the FZ and Stage I to Stage IV radula regions, consistent with coordinated gene expression changes during radula development. While more genes were found downregulated (1,078) than upregulated (301), comparison of genes differentially expressed at each stage identified those with roles in chitin and iron metabolism that are clearly relevant for limpet tooth formation. Transcripts encoding chitin synthases (CHS) were expressed along the radula, with specific genes having differential expression patterns and Stage I with the highest chitin synthase expression of any stage. Observations that the bulk of chitin synthesis occurs in the early radula stages are consistent with the spatiotemporal generation of chitin scaffolds. However, the differential expression of specific CHS genes at specific stages along the radula suggests unique functional requirements at each stage of development.

Of the three chitinase 1 (CHIT1) transcripts, two were highly expressed in the FZ with expression decreasing along the radula stages with the third transcript (DN209778) expressed at high levels all along the radula, except for Stage I. The high expression levels of chitinases concomitant with several chitin synthases in the FZ may represent a high level of plasticity and remodelling of chitin at this stage. The concomitant increase in chitin synthase and decrease in chitinase 1 expressions in Stage I ought to correspond to a net increase in chitin production. Interestingly, CHIT1 transcript DN209778 was also expressed in mineral-containing Stages II-IV, suggesting continued chitin remodelling, possibly to accommodate goethite crystal intercalation.

The differential gene expression profiles along the radula length from the FZ to Stage IV show a consistent pattern, where reductase expression is increased and oxidase expression is decreased, presumably as teeth accumulate increasing amounts of ferric iron ($Fe^{3+}$), in goethite. Specifically, the non-mineralised Stage I radula expresses the melanotransferrin (TRFM), which binds ferric iron ($Fe^{3+}$)[33]. Stages I and II express a transcript for heavy metal binding protein HIP, which has been previously described as a carrier of divalent cations[34] but may also bind to class A, B and borderline metals[35], which include both $Fe^{2+}$ and $Fe^{3+}$. Importantly, ascending gradients of expression, from the FZ to Stage IV are also found for the divalent metal transporter Cyclin and CBS domain divalent metal cation transport mediator 2 (CNNM2), and for enzymes associated with the reduction of $Fe^{3+}$ to $Fe^{2+}$. These include two transcripts for Ferric-Chelate Reductase (DN211629 and DN211756) and for the metalloreductase six-transmembrane epithelial antigen of

prostate 3 (STEAP3). Moreover, a complementary descending gradient of gene expression from the FZ to Stage IV occurs for Hephaestin-like 1 (HEPHL1), protein which acts as an iron transporter and a ferroxidase that oxidises $Fe^{2+}$ to $Fe^{3+}$[36].

Hilgers et al. identified transcripts differentially expressed between the foot, mantle and radula of the freshwater snail *T. sarasinorum*[17]. We identified gene orthologs (Supplementary Fig. 1e) with high levels of conservation. These included developmental genes such as Aristaless-related homeobox protein (ARX) and Gastrulation Brain Homeobox gene (GBX), which in our data were found expressed in the FZ, and Stages I-IV respectively, suggesting their involvement in a potentially important patterning 'switch' in radula generation. The Hes Family BHLH Transcription Factor 1 (HES1) has been associated with stem cells across mammals and invertebrates[37,38], which agrees with its expression in the FZ, where cells isolated from this region showed clear stem cell potential. *HES1* has also been shown to function as a transcriptional repressor, which may explain its up-regulation in Stage IV radula[39]. Also, specifically up-regulated in the FZ was the chitinase Di-N-acetylchitobiase (DIAC), which could be important in chitin restructuring for limpet tooth formation.

Secreted protein, acidic, rich in cysteine (SPARC), associated with mineralisation across a range of organisms[17], was found expressed in Stages I-IV of the radula. The expression of superoxide dismutase (SODC) was similarly up-regulated between Stage I and Stage IV radula.

In natural aquatic environments the superoxide anion acts as a reductant, while the radical acts as an oxidant and, as such, it has the potential to switch iron between water-soluble $Fe^{2+}$ and non-soluble $Fe^{3+}$, and the latter may find its way into cells[34]. Thus, the superoxide radical may affect iron solubility and bioavailability[40], and the presence of SODC, which functions to remove superoxides, may be important for stable iron deposition within the limpet teeth. The conserved transcription factor ets1, previously identified in the radula of *T. sarasinorum*, we found specifically up-regulated in Stage III, supporting a role in radula development. The gene encoding Protein PIF was up-regulated specifically in Stage IV of the radula. PIF in the nacre of shells contributes to the matrix required for aragonite crystal formation[23], so it might perform a similar role in ECM binding and goethite crystal formation in the mature limpet tooth.

The extremely powerful regeneration and organogenesis potential of the FZ suggests that it is central to directing the activity and organisation of surrounding cells. This effect is comparable to previous studies in sponges, whereby completely separated cells can re-aggregate to form a new sponge structure[41]. However, sponges are basal metazoans and lack the complex arrangement of tissues and organs found in gastropods, which highlights the exceptional nature of this self-assembly of the limpet radula cells.

In summary, using a combination of molecular and functional approaches, we have identified the mechanism behind the formation of the tooth of the Common Limpet, made of the

strongest known biologically occurring material[5]. We applied this knowledge to develop novel organotypic and cell culture methods that allowed maintenance of highly specialised cells to the stage where they evoked organogenesis of de novo radula structures and even of individual teeth fully resembling those found in vivo. Moreover, we assembled an acellular set of key components, which enabled us to produce, in a biomimetic fashion, the limpet-tooth material under laboratory conditions. Our data indicate the presence of iron in radula cells at the point of harvesting, which was in a sufficient quantity to allow subsequent low energy generation of mineral mediated by factors present in limpet cell-conditioned media. Interestingly, the Raman spectra were consistent with hematite ($Fe_2O_3$) crystals in this biocomposite. The presence of hematite rather than goethite is intriguing; both goethite and haematite contain ferric iron ($Fe^{3+}$) and it is already established that under certain conditions goethite may transform to haematite[42,43].

The proof-of-concept presented in this study can be scaled up using made-to-measure chitin sheets and synthetic substitutes for limpet cell-conditioned media. Given that chitin is currently a waste by-product of the fishing industry[44], our approach would allow its repurposing into a novel composite material that could substitute for many existing synthetic materials that are manufactured in a polluting or unsustainable manner, and could help solve environmental challenges such as the ocean plastics crisis. Furthermore, as chitin is itself biodegradable, this bioinspired composite meets the key modern engineering challenge of sustainability. In short, this new material has the potential to be manufactured and disposed of without generating harmful waste products.

## Methods

**Cell isolation and culture**. Common limpets (*Patella vulgata*) were harvested locally (Portsmouth, UK) under the University Animal Welfare and Ethical Review Body (AWERB) approval (821B). Individual limpets were treated as biological replicates and, where possible, multiple analyses were undertaken in each individual to minimise the number. Dissections were carried out using a Zeiss Stemi 305 compact stereo microscope under a sterile laminar flow cabinet. All cell culture reagents were from Sigma-Aldrich (St. Louis, Missouri, USA) and tissue culture-treated plastic ware from Thermo-Fisher (Waltham, Massachusetts, USA) unless otherwise stated. Artificial Seawater (ASW) was made following the recipe from Cold Spring Harbour protocols[45]. L15-ASW was made from L15 media with phenol red (Thermo-Fisher), 400 mM NaCl, 10 mM KCl, 15 mM HEPES, 10 mM $CaCl_2$, 55 mM $MgCl_2$, 0.4 mM L-glutamine (Thermo-Fisher)[30,46]. Both ASW and L15-ASW had additional gentamicin (250 µg/mL), penicillin-streptomycin (Thermo-Fisher) (100 U/mL), Amphotericin B (250 ng/mL), Diuron (10 ng/mL) and $CuSO_4$ (6.3 µM).

All media were adjusted to pH 7.8–7.9 (corresponding to the pH of limpet hemolymph) and sterile filtered through a 500 mL volume Filtropur V50 with 0.1 µm pore size (Sarstedt, Nümbrecht, Germany) prior to use.

For organotypic cultures, 36 isolated Formation Zones (FZ) were studied in 6 independent experiments. Dissected FZ were incubated immediately in 24 well plates containing 1 mL/well of L15-ASW. Organotypic cultures were given 50% media changes on days 2 and 7. Twelve FZ were used in experiments with iron sulphate supplementation, where fresh media containing between 400 nM and 2 µM Fe(II)$SO_4$ were applied.

For cell isolation, whole radulae or individual sections of the radula (the FZ and Stages I to IV) were mechanically dissociated before 18-hour incubation in Dispase II (10 U/mL) in L15-ASW on a shaker at room temperature. After this incubation step, DNase I (1 U/mL) (Thermo-Fisher) was added to prevent clumping prior to the trituration of suspensions. Dispase II and DNase I were inhibited with 1 mM EDTA and suspensions were filtered through 50 µm sterile cell strainers before centrifugation at 400 g for 5 min and resuspension in 1 mL L15-ASW. Cells were counted on C-Chip haemocytometers (LabTech, East Sussex, UK) prior to seeding on either glass coverslips, or tissue culture plastics.

5 mm glass coverslips (manufactured by Knittel glass, Braunschweig, GERMANY; supplied by Richardsons of Leicester, UK) and 32 mm large glass cover discs (Chance Glass, Malvern, UK) were sterilised prior to use in an oven at 200 °C for 2 h. Discs of electrospun chitin were cut with an integra sterile 6 mm biopsy punch (VWR, Radnor, Pennsylvania, USA) and sterilised under UV light on both sides prior to use.

5 mm glass coverslips and 5 mm discs of electrospun chitin were seeded in L15-ASW at a density of 50,000 cells each in 96 well plates. For harvesting, samples

were either fixed in formalin-ASW for staining or for enzymatic assays, cultures grown on glass were rinsed in ASW then lysed in 200 µl/well of $dH_2O$ and snap frozen at −80 °C. 32 mm large glass cover discs were seeded with $1–3 \times 10^6$ each in 6 well plates or 35 mm petri dishes. For conditioned media collection, 6 well plates were seeded with $1 \times 10^6$ cells from stage III and IV radula per well.

Cultures were maintained in a Heratherm Refrigerated Incubator IMP180 (Thermo-Fisher) at 14 °C for up to six weeks. All cultures were repeated with at least three independent replicates.

**Transcriptome profiling**. For the purposes of RNA sequencing, radulae were isolated from ten similar sized female limpets with a shell diameter across the widest point of 40–48 mm, and radula length of 75–87 mm, and the five with the highest quality RNA extracted across the different tissues were used for further studies. Five replicates per condition was predicted to be sufficient for ~90% power to detect changes of 2-fold with an alpha threshold of 0.05, based upon gene expression variance estimates from RNA sequencing of 5 oyster trochophores (see http://www.oysterdb.com).

Radulae were dissected from limpets under a Zeiss Stemi 305 compact stereo microscope and divided into five sections (the FZ, Stages I, II, III and IV)[47]. A sample of foot muscle was dissected from each limpet as a control for endogenous limpet genes. All samples were immediately snap-frozen in 500 µl QIAzol and total RNA extracted with the RNeasy Mini Kit (Qiagen, Germany) following manufacturer's guidelines. The amount of RNA present was quantified with a Nanodrop (Thermofisher, USA) and integrity was assessed using the Nanodrop and Agilent Bioanalyzer 2100 (Agilent, USA). RNA integrity scores ranged from 3.8 to 8 (mean = 6.3 ± 0.9; Supplementary Table 1).

*Next-generation sequencing.* NGS libraries were created for all 30 samples using the TruSeq Stranded mRNA library prep kit (Illumina, California, USA), and were multiplexed and sequenced by Theragen Etex, South Korea (theragenetex.com) using the Illumina HiSeq 2500 sequencer (Reagent Kit v1.5; HiSeq Control Software v2.2.58) to produce paired-end 100 bp sequences. Reads were processed and assembled to create a complete transcriptome assembly as described in more detail in the Nature Research Reporting Summary for this paper. Transcript abundance was estimated for individual libraries against the assembled transcriptome using Kallisto v0.43.1[48] with parameters "–rf-stranded". The abundance of transcripts for each sample based on fragments per kilobase mapped (FPKM) can be seen in Supplementary Data 2. Transcripts were filtered to keep only those greater than 500 bp in length with an identified open reading frame (ORF) and a FPKM abundance score above 1 in at least one sample. In addition, transcripts most closely matching Archaeal, Bacterial or Viral species were removed. A workflow of the filtering process can be seen in Supplementary Fig. 1a. Differential expression analysis between the different tissues was conducted using the DESeq2[49] package in R[50]. The results from the comparison of the radula sections with the muscle samples using DESeq2 for all transcripts can be seen in Supplementary Data 3. Resulting $p$ values were adjusted for multiple testing using the Benjamini and Hochberg false discovery rate correction[51]. Genes were identified as differentially expressed between tissues based on a fold-change greater than 2-fold (upregulated or downregulated), with an adjusted $p$ value less than 0.05. In addition, differentially expressed genes with a mean FPKM below 1 for both tissues were removed to avoid over-inflating changes in low-abundance transcripts. Additional annotation was performed against the Protein family (Pfam) database[52] (v31.0) using HMMER[53] (v3.1b2), Clusters of Orthologous Groups of proteins (eggNOG) database[54], the Kyoto Encyclopaedia of Genes and Genomes (KEGG) database[55], and Gene Ontology (GO) database[56]. Results were collated into a single output table using Trinotate v3.02 (http://trinotate.github.io/). The annotation of unique transcripts from the assembled transcriptome can be seen in Supplementary Data 1.

*Quantitative PCR.* Quantitative PCR (qPCR) was carried out to confirm differential expression of key genes associated with limpet tooth production, and normalised to putative housekeeping genes identified by NGS. RNA was isolated as described, and cDNA generated using SuperScript IV VILO Master Mix (Thermofisher) according to manufacturer's instructions. Primers were designed to target genes using NCBI Primer-BLAST with target product sizes of 150–220 bp, primer melting temperatures of 59–61 °C, primer size of 19–21 bp and primer GC content of 50–60% (Supplementary Table 2). Primer specificity was validated by end-point PCR with GoTaq Green master mix (Promega) on a Primus 96 Plus Thermal Cycler with the following cycle parameters: Initial denaturation 2 min at 95 °C, followed by 35 cycles of 1 min denaturation at 95 °C, 1 min annealing at 57 °C and 1 min extension at 73 °C prior to a final 5 min extension at 72 °C and soak at 4 °C. End-point PCR products were ran out by gel electrophoresis in 1% agarose with 5 µg/mL ethidium bromide. Gels were visualised on a GelDoc EZ Imaging System (Bio-Rad).

qPCRs were carried out with PrecisionPLUS qPCR Master Mix with low ROX and SYBRgreen (Primer Design) on an Applied Biosystems ViiA 7 Real-Time PCR System with the following cycle parameters: 2-minute hot start at 95 °C, followed by 45 cycles of 10 s denaturation at 95 °C, and 1-minute data collection at 60 °C. Data were normalised to housekeeping genes (RLA0, RS13) using the 2-ΔΔ Ct method.

**X-ray microcomputed tomography and analysis**. Wholemount limpets were air-dried, while isolated radulae and organogenic cultures were fixed in formalin in ASW and transferred to 70% ethanol prior to scanning. Samples were imaged using high-resolution X-ray computed tomography (XCT) (ZEISS Xradia Versa 520 Versa, Carl Zeiss X-ray Microscopy, Pleasanton, CA, USA). The XCT system was set to operate at 50 kV and 4 W for all samples. Isotropic voxel sizes ranged from 1.5 to 9.83 µm. For all samples, projection images were acquired over 360° at equal intervals. The projection images were then reconstructed using the manufacturer's integrated software (Scout and Scan Reconstructor, Carl Zeiss X-ray Microscopy, Pleasanton, CA, USA), which utilises a filtered back-projection reconstruction algorithm. The reconstructed tomograms, 16-bit grey-level images, were analysed using DragonFly Pro (Object Research Systems, Quebec, Canada) for image segmentation. Visualisation of the reconstructed volumes was carried out using TXM3DViewer (Carl Zeiss X-ray Microscopy, Pleasanton, CA, USA). Figure 1i was rendered with VGStudio 2.2 (Volume Graphics, Germany).

**Cell functional assays**. Cell viability was quantified using the Presto Blue assay (Thermo-Fisher) by adding 10 µl of reagent to 90 µl L15-ASW per well in a 96 well plate. Optical density was recorded 7 and 14 days later at 570 nm on a POLARstar Optima plate reader (BMG labtech, Ortenberg, GERMANY).

Iron deposition was visualised from fixed samples by staining for 20 min with Prussian blue counterstained with Nuclear Fast Red to reveal cells and visualised by light microscopy.

Additional functional assays were performed using a Spectra Max i3x Multi-Mode microplate reader (Molecular Devices, San Jose, California, United States) on lysates from tissue harvested directly from the limpet and from cells maintained in culture.

Chitin was quantified from by air-drying 20 µl of cell lysate in a 96 well plate and incubating with Calcofluor White Stain (1 g/L Calcofluor White M2R, 0.5 g/L Evans blue) for 5 min. Wells were rinsed with $dH_2O$ and fluorescence read at excitation 355 nm, emission 433 nm. Standards were generated with a suspension of chitin from shrimp shells (Sigma Aldrich - Lot # SLBR6796V).

Chitin synthase activity was quantified by incubating 20 µl cell lysate with 30 µl $dH_2O$ and 50 µl of chitin synthase substrate (10 mM *N*-acetyl glucosamine, final concentration 5 mM), for 1 h at 37 °C. Negative controls were prepared with paired samples containing 20 µl cell lysate and 80 µl $dH_2O$. Chitin was quantified using Calcofluor white as described and the difference between samples with and without *N*-acetyl glucosamine used as an indicator of net chitin synthase/chitinase activity.

Chitinase activity was quantified from cell lysates with a fluorimetric Chitinase Assay Kit following manufacturer's instructions (CS1030, Sigma Aldrich) and read at excitation 360 nm, emission 450 nm.

Total iron ($Fe^{2+}$ and $Fe^{3+}$) was quantified following the protocol described by Hedayati et al. (2017)[57]. Standards were generated from 100 mM Iron Standard (#MAK025D, Sigma-Aldrich).

Data from chitin, chitin synthase, chitinase, ferrous iron, total iron and ferric iron assays was standardised to protein content using the bicinchoninic acid (BCA) assay (Thermo-Fisher) following manufacturer's instructions.

All functional assays were repeated with at least three independent replicates.

**Mineral preparation for X-ray microanalysis**. Control goethite (Sigma, 71063) and mineral isolated from the matrix surrounding limpet radula cells in vitro were suspended in deionized water and dropped on Formvar/Carbon 300 Mesh Ni grids (Agar Scientific Ltd.). Samples were dried for 30 min at room temperature. The grids were counterstained with 2.5% uranyl acetate in ethanol for 20 min at room temperature in the dark. Grids were washed sequentially in 50% ethanol, deionized water then dried. Samples were observed using a JEM-1400 transmission electron microscope (TEM, JEOL Co.) with adapter for X-ray microanalysis INCA X-sight 7215 Energy Dispersive Spectrometer (EDS, Oxford Instruments) and MORADA CCD high-resolution digital camera (SiS-Olympus)[58].

**Electrospun chitin**. The very limited solubility of Chitin in organic solvents was increased by ultrasonication in aqueous suspension, a process that can finely control the partial depolymerisation process (in contrast to a previous study which relied on gamma irradiation[29]). A suspension of chitin from shrimp shells (Sigma Aldrich - Lot # SLBR6796V) in deionised water (15 g/L; pH 6) was subjected to ultrasonication (Ultrasonic Processor VCX 600-5 equipped with a CV26 ½" probe; 30 min) followed by flash freezing in liquid nitrogen and freeze-drying using a VirTis BT4kZL instrument.

Chitin fibres were obtained by electrospraying a 20% w/w solution of the above chitin in 1,1,1,3,3,3-hexafluoroisopropanol (HFIP) using a Spraybase® Projector (Life Sciences Ltd, Ireland) equipped with a high-voltage supply source and a software-controlled programmable syringe pump (World Precision Instruments, Florida, US). The process was conducted in stable 'cone-jet' mode at a rate of 10 mL/h, using a potential voltage in the range 9.2–10.8 kV and a tip-collector distance of 11 cm. All experiments were carried out at ambient temperature (23 °C). The chitin fibres were collected dry on aluminium foil plates.

**Scanning electron microscopy**. Electrospun chitin samples (including those mineralised by limpet cell-conditioned media) were dried at critical point with liquid $CO_2$ using a Leica EM CPD300, then mounted on carbon adhesive tabs and sputter coated using a gold-palladium target in a Quorum 150 Coating Unit. Specimens were imaged by scanning electron microscopy using a Tescan Mira 3 with a Schottky field emission gun operated in high vacuum mode at an acceleration voltage of 15 kV. Images were captured at a working distance of 15 mm in secondary electron mode. Elemental composition of particles found on the electrospun chitin was examined by energy dispersive X-ray spectroscopy (EDS) using an Oxford Instruments XMax 50 EDS detector and INCA software (version 5.05) to map elemental distribution.

**Raman spectroscopy**. A Renishaw inVia Qontor Raman microscope was used for the spectroscopy. Samples were analysed using an air cooled 300 mW 785 nm laser at 1% power under 50x magnification for 5 accumulations at 10 s exposure. Areas of iron accumulation were identified using Prussian blue staining on chitin scaffolds. Raman data were collected from areas adjacent to the most intense staining. Spectral libraries used were the L60000 complete ST Japan library (Sept 2020, 16898 spectra) and the Renishaw Minerals and Inorganic materials library (Sept 2020, 1000 spectra).

**Atomic force microscopy**. Chitin samples, along with glass, polypropylene (PP) and poly(methylpentene) (PMP) reference materials[59] were attached to nickel stubs prior to AFM analysis using a Multi-Mode/NanoScope IV scanning probe microscope (Bruker, Santa Barbara, CA, USA)[60].

A single silicon probe ($t = 3.6–5.6$ µm, $l = 140–180$ µm, $w = 48–52$ µm, $v_0 = 312.73$ kHz, $k = 12–103$ N m$^{-1}$, $R < 7$ nm; model: OTESPA, Bruker, France) was used for all the AFM measurements. Force *vs.* distance curves were obtained 1000 nm away from the surface to a 100 nm cantilever deflection trigger against the samples. Approach and retraction force curves (100 each) were obtained in a $10 \times 10$ area, each point separated by 1000 nm (approach and retraction speed = 3.69 µm s$^{-1}$) was used. The sensor response (47.02 nm V$^{-1}$) was obtained on the glass surface and no subsequent AFM laser nor photodetector repositioning was made.

Typical approach force curves ($n = 5$) obtained from each material were converted to text files using NanoScope Analysis software (V 1.4, Bruker). Force curves from the chitin, PP and PMP materials were aligned with those from the glass (hard reference) substrate to achieve a common tip-sample contact point. Reduced moduli were obtained from the linear portion of force vs. indentation$^{3/2}$ plots according to Hertzian mechanics[61], modelling the AFM tip as a spherical indenter[59]. Young's moduli were calculated from the reduced moduli assuming a Poisson ratio of 0.5[59]

**Statistics**. Statistical tests were carried out in GraphPad Prism 8 and IBM SPSS Statistics 25 software packages. Shapiro Wilk and Kolmogorov-Smirnov normality tests and ROUT outliers tests were carried out in GraphPad Prism, and showed data were approximately normal. Where more than two groups were compared, significance was determined using a Univariate Analysis of Variance and differences between individual treatment groups were determined using either Tukey or Dunnett's post hoc tests as appropriate. Figures were generated in GraphPad Prism 8 and data presented with individual data points and mean with standard deviation error bars.

**Figure assembly**. Final figures were assembled using the FigureJ plugin in ImageJ[62].

**Reporting Summary**. Further information on research design is available in the Nature Research Reporting Summary linked to this article.

## Data availability

All raw sequence read level data analysed during this study are available from the NCBI Sequence Read Archive (SRA) under Bioproject PRJNA566106. The transcriptome assembly is available from the NCBI Transcriptome Shotgun Assembly Archive (TSA) under accession GHWI00000000. The full annotation for the assembled transcriptome has been deposited in FigShare under a CC0 license under https://doi.org/10.6084/m9.figshare.15035988. Transcripts were annotated against the Universal Protein Knowledge Base (UniProtKB) curated SwissProt database release 2018_01 (https://www.uniprot.org/uniprot/?query=reviewed:yes) and the Protein family (Pfam) database version 31.0 (https://pfam.xfam.org/). Spectral libraries used for Raman microscopy were the L60000 complete ST Japan library (Sept 2020, 16898 spectra) and the Renishaw Minerals and Inorganic materials library (Sept 2020, 1000 spectra) All other data generated or analysed during this study are included in this published article (and its Supplementary information files).

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

## Acknowledgements

Funding from the University of Portsmouth Themes Research and Innovation Fund (TRIF) is gratefully acknowledged. The authors acknowledge The Zeiss Global Centre (University of Portsmouth) for providing XCT imaging and image reconstruction to conduct this study. S.C.R. is part funded from Research England's Expanding Excellence in England (E3) Fund. We also thank Joe Dunlop for assistance with SEM and EDS, and Marc Martin for video editing Supplementary Movies 1 and 3.

## Author contributions

R.R. carried out all limpet cell culture, functional assays, staining and molecular biology. D.C.G., A.T.F. and A.H.B. designed the project and secured initial funding while D.C.G., A.T.F., R.R. and M.S. developed the project and secured further funding. S.C.R. carried out bioinformatics analysis of the radula transcriptome and upload of raw and processed data to public repositories. A.K. carried out XCT scans and analysis with support from R.R., R.P. and G.T. L.B. carried out TEM EDX on crystals isolated from limpet cells in vitro. E.B. designed protocols for solubilisation and electrospinning of chitin. R.R. conceived methods for mineralising chitin with limpet cell conditioned media. S.M.C. carried out SEM and EDS on mineralised chitin scaffold. F.S. carried out Raman spectroscopy, J.R.S. carried out the AFM. R.R. and D.C.G. wrote the manuscript with assistance from all other co-authors.

## Competing interests

The authors declare no competing interests.
