## [Peer Review File · Nature Communications]

Title: Biomimetic generation of the strongest known biomaterial found in limpet toothREVIEWER COMMENTS

Reviewer #1 (Remarks to the Author):

This interesting paper describes a novel phenomenon that cells from certain regions of tissues in the radula of limpets can form a neo-small organ in vitro and more surprisingly, it can mineralize chitin fibers in vitro with enhanced mechanical properties. To provide insight into the formation mechanism, the authors used RNA-seq of different regions of the radula and by differential gene analysis, they identified some key genes regulating the teeth formation.

Overall, this paper is innovative in terms of methods and is significant for the biomaterial field. However, the data especially the transcriptome is not fully analyzed and compared to the existing literature; the micro-CT image is not demonstrated clearly. In addition, the final material synthesized is mineralized chitin fibers but not the limpet teeth-like structure. So “the strongest biomaterial” claim in the title may be too strong.

I have the following comments:

Major:

1. Regarding the transcriptome (table 3), I suggest the author to dig deep into this data. For example, Pif protein, a VWA and chitin-binding protein, has been shown to play important roles in the formation of shells (Michio Suzuki et al, Science, 2009), and has also been found in the limpet teeth (Chuang et al, Proteomics, 2018). This protein is repeatedly found to be highly expressed in all stages of radula. However, the author did not analyze it at all. Some other interesting examples are SOD, EGF proteins, Chitin-binding proteins. This in-depth analysis will provide insight into the formation of the tooth.

2. P37: “the processes which underlie its formation have remained unclear” This is not true. There are already some literature tackling this problem by RNA-seq, proteomics or the combined methods. These should be mentioned and compared in the introduction and/or discussion.

See references:

1) Leon Hilgers et al, Novel Genes, Ancient Genes, and Gene Co-Option Contributed to the Genetic Basis of the Radula, a Molluscan Innovation, Molecular Biology and Evolution, 2018

2) Chuang Liu et al, The Microstructure, Proteomics and Crystallization of the Limpet Teeth, Proteomics, 2018

3) Michiko Nemoto et al, Integrated transcriptomic and proteomic analyses of a molecular mechanism of radular teeth biomineralization in *Cryptochiton stelleri*, Sci Rep, 2019

3. “While the composition and structure of LT has been well characterized,” the author should at least briefly describe the major findings before.

4. The final product in vitro is haematite (Fe₂O₃) instead of goethite in the limpet teeth. Can the authors discuss this more in-depth? I noticed that the author used Fe²⁺ instead of Fe³⁺ in the precursor, why?

5. The author claimed they produced mineralized chitin by the conditioned medium of radula in figure

4c-f. However, there is only one mineral particle in the image, can the author provide a larger view of this image?

Minor:

1. P34: "Chiton" should be "chiton"
2. I suggest the authors first provide a limpet image and then a whole radula followed by dissected radula for the readers that are not familiar with the system.
3. The micro-CT image is not clearly shown for better indicating the different stages of mineralization. I even suggest moving this image into the main Figures instead of putting into the supplemental materials. Similarly, the micro-CT image in supplemental figure 2 is not clear at all.
4. What is the difference between figure 1g and figure 1h?
5. Which part are the head and tail respectively in figure 1 q-s?
6. I suggest to move table 1 into supplemental materials because it is very long and does not contain vital information. Actually, table 3 contains many interesting and important information regarding the formation of the tooth. I suggest the author say more about table 3.
7. P192-194: "The lack of chitinase activity in FZ, concomitant with high expression of 193 different chitinase genes there may suggest a lack of translation, or the generation of 194 protein precursors requiring activation". In the reference I mentioned by Chuang et al, they used proteomics rather than only transcriptome. In their results, they did not find any chitinase in the mature tooth neither. This may reflect that chitinase is only needed between stage II to III. And actually chitin-binding proteins may play important roles in the formation of the tooth.
8. About the qPCR results, is the y-axis the relative fold compared to that in the foot tissue?

Reviewer #2 (Remarks to the Author):

This manuscript is devoted to the molecular basis of the formation and biomineralization of radular teeth in true limpet *Patella vulgata*. Despite vast data of gastropod radula morphology and functioning, there are no study of a molecular basis of the radula biomineralization. This very professional study was done at a high level and combined various methods entirely adequate for the tasks. New data about genes expression in different radula zones involved in the synthesis and mineralization of the radula were obtained. Mapping of the radular zones and the ratio expression of key genes is a valuated contribution to understanding the formation of the chitinous matrix and the processes of biomineralization.

A very significant result of this study is that the authors cultured the cells of the radular sac and, for the first time, synthesized individual chitinous teeth outside of the organism and biomineralized chitinous scaffold in vitro. These new technologies could give a stimulus to a whole direction in the production of biomimetic materials.

I have a few comments:

1. St 48. Since there are no clear definitions of the terms "stages" and "Formation Zone" many questions arise, for example, what is the difference between the formation zone and stage I? If teeth are formed

in the formation zone, what happens in stage I? What happens with teeth in the maturation zone (stage II)? Why do you use the term zone for the formation zone and stage for the subsequent zones? It would be nice to imagine the processes that occur in different stages / zones to explain the difference in gene expression. Understanding the processes occurring in different stages / zones can be the basis for explaining the difference in expression.

In material and methods (st 474), you refer to Mann 1986: do you identify stages based on his data on tooth mineralization?

Would you please add a full description of the stages?

1. I didn't find supplementary figures 1-2; you probably mean video 1-2?

2. St. 72. Since a whole radular sac was taken for the cell culture, all cells that synthesize the teeth and radular membrane, including odontoblasts, membranoblasts and surrounding epithelia, were transferred to this culture. Thus, the normal tissue composition and tissue interactions, which allow radula to be synthesized, are preserved. In this case, tooth formation in the culture testify about normal functioning of the radular sac interactions more than about the pluripotency. If a structure not characteristic of the radula were formed in this cell culture, then it would be pluripotency. The fact that these cells retain the ability to synthesize a tooth in culture, outside of such a complexly organized structure as the radular sac, is, in my opinion, much more surprising than pluripotency. I suggest adding a paragraph about this.

3. St 98. What do you mean when you write radula? All radular sack, including Formation Zone? Please specify.

4. Both sup figure 5-6 named in the site as fig 4.

5. It is important to add a total number of studied specimens to material and methods.

6. Decipher all abbreviations across the manuscript (for example, BCA, st. 569).

REPLY TO REVIEWER COMMENTS

We would like to express our thanks to both Reviewers for their very positive comments on our manuscript and even more for their constructive and insightful suggestions. Reviewers identified areas where we could improve our manuscript and we performed the additional analyses suggested and added these results in the revised version of the manuscript.

Reviewer #1 (Remarks to the Author):

This interesting paper describes a novel phenomenon that cells from certain regions of tissues in the radula of limpets can form a neo-small organ in vitro and more surprisingly, it can mineralize chitin fibers in vitro with enhanced mechanical properties. To provide insight into the formation mechanism, the authors used RNA-seq of different regions of the radula and by differential gene analysis, they identified some key genes regulating the teeth formation.

Overall, this paper is innovative in terms of methods and is significant for the biomaterial field. However, the data especially the transcriptome is not fully analyzed and compared to the existing literature; the micro-CT image is not demonstrated clearly. In addition, the final material synthesized is mineralized chitin fibers but not the limpet teeth-like structure. So "the strongest biomaterial" claim in the title may be too strong.

I have the following comments:

Major:

1. Regarding the transcriptome (table 3), I suggest the author to dig deep into this data. For example, Pif protein, a VWA and chitin-binding protein, has been shown to play important roles in the formation of shells (Michio Suzuki et al, Science, 2009), and has also been found in the limpet teeth (Chuang et al, Proteomics, 2018). This protein is repeatedly found to be highly expressed in all stages of radula. However, the author did not analyze it at all. Some other interesting examples are SOD, EGF proteins, Chitin-binding proteins. This in-depth analysis will provide insight into the formation of the tooth.

We are very pleased with this positive evaluation and thank the Reviewer for the constructive comments. In response to this specific feedback, we have performed additional analyses of the transcriptome data. Specifically, we analysed the expression of target genes highlighted by the Reviewer and expanded this analysis to include other candidate genes from recently published papers. The differential expression is demonstrated in Supp. Fig 2e. Although VWA transcripts were identified, none had a higher expression in the FZ or radula compared to muscle. However, the additional analysis did reveal differential expression along the FZ and radula of DIAC, ARX, SPRC, SODC, GBX2, ETS1, PIF and HES1 genes.

Furthermore, this suggested in-depth analysis provided a further important insight into the tooth formation in terms of the identification of differentially expressed genes found to be involved in vesicular transport and secretion associated with mineralisation of chitin scaffolds (Supplementary Figure 4).

In the 'Results' section under the sub-heading 'Differential gene expression reveals functional specialisation across the radula', we have added the following text:

Moreover, previous studies have identified genes with higher expression in the radula than in muscle or shell mantle ^{17,18,22}. Differential expression of such candidate genes across radula stages was analysed in depth (Supplementary Fig. 1e). DIAC and ARX expressions were specifically enriched in FZ only. SPRC, SODC, GBX2 and FRIS were found from SI to SIV radula, while ETS1 appeared up-regulated at SIII. There are two subsets of PIF transcript; one subset (PIF_1) is more highly expressed in SIV radula than elsewhere, while another (PIF_2) is predominantly expressed in muscle. HES1 was also markedly upregulated at Stage IV. Other genes, including those encoding chitin-binding proteins (other than PIF-1) were not differentially expressed across the radula.

In the discussion, we have added the following:

"Hilgers *et al* identified transcripts differentially expressed between the foot, mantle and radula of the freshwater snail *Tylomelania sarasinorum* ¹⁷. We identified and compared gene orthologs (Supplementary Figure 2c) with high levels of conservation. These included developmental genes such as Aristaless-related homeobox protein (*arx*) and Gastrulation Brain Homeobox gene (*gbx*), which in our data were found expressed in the FZ, and stages I-IV respectively, suggesting their involvement in a potentially important patterning 'switch' in radula generation. The Hes Family BHLH Transcription Factor 1 (*Hes1*) has been associated with stem cells across mammals and invertebrates ^{31,32}, which agrees with its expression in FZ, where cells isolated from this region showed clear stem cell potential. *Hes1* has also been shown to function as a transcriptional repressor, which may explain its up-regulation in Stage IV radula ³³. Also, specifically up-regulated in the FZ was the chitinase Di-N-acetylchitobiase (*diac*), which could be important in chitin restructuring for limpet tooth formation.

Secreted protein, acidic, rich in cysteine (*sprc*), associated with mineralisation across a range of organisms ¹⁷, was found expressed in Stages I-IV of the radula. The expression of superoxide

dismutase (*sodC*) was similarly up-regulated between SI-SIV radula. The superoxide radical may affect iron solubility and bio-availability³⁴, therefore the presence of *sodC* may be important for stable iron deposition within the limpet teeth. The conserved transcription factor *ets1*, previously identified in the radula of *T.sarasinorum*, we found specifically up-regulated in Stage III, supporting a role in radula development. The gene encoding Protein PIF was up-regulated specifically in Stage IV of the radula. PIF in the nacre of shells contributes to the matrix required for aragonite crystal formation²², so it might perform a similar role in ECM binding and goethite crystal formation in the mature limpet tooth.”

2. P37: “the processes which underlie its formation have remained unclear” This is not true. There are already some literature tackling this problem by RNA-seq, proteomics or the combined methods. These should be mentioned and compared in the introduction and/or discussion.

See references:

- 1) Leon Hilgers et al, Novel Genes, Ancient Genes, and Gene Co-Option Contributed to the Genetic Basis of the Radula, a Molluscan Innovation, *Molecular Biology and Evolution*, 2018
- 2) Chuang Liu et al, The Microstructure, Proteomics and Crystallization of the Limpet Teeth, *Proteomics*, 2018
- 3) Michiko Nemoto et al, Integrated transcriptomic and proteomic analyses of a molecular mechanism of radular teeth biomineralization in *Cryptochiton stelleri*, *Sci Rep*, 2019

We thank the Reviewer for pointing out these publications and we have used the candidate genes identified in these papers in the aforementioned in-depth analyses of the transcriptomic data. We also addressed this oversight by modifying the Introduction, which now reads as follows:

“The composition and structure of limpet tooth material have been studied for over a century. Specifically, the presence of a chitin scaffold was established in 1907¹⁶; goethite as a key constituent was identified in the 1960’s⁶ and directionally arranged nanofibrous crystals of goethite within a highly organised chitin matrix composing each tooth were described in 1980’s⁷. However, it is only recently that the processes which underlie limpet tooth formation and are critically important for the development of biomimetics, could be unravelled using molecular approaches.

RNA-seq has shown that the radula transcriptome from the freshwater snail *Tylomelania sarasinorum* contains groups of genes associated with vesicular secretion, chitin binding and iron transport¹⁷. Proteomics on radulae from the limpet *Cellana toreuma* revealed the

presence of ferritins in the teeth, while GTPases were identified as the predominant goethite binding proteins¹⁸. A detailed study on the chiton species *Cryptochiton stelleri* characterised deposition of ferrihydrite in the tooth cusp, which transforms to magnetite between just a few rows of teeth¹⁵.

As the radula is structured as a conveyor belt for development, microdissection combined with molecular analyses can reveal the step-wise formation of limpet teeth, including chitin scaffold formation and gradual inclusion of goethite crystals⁷.”

3. “While the composition and structure of LT has been well characterized,” the author should at least briefly describe the major findings before.

We have addressed this with the changes to the Introduction in response to the Major Comment 2, as explained above.

4. The final product in vitro is haematite (Fe₂O₃) instead of goethite in the limpet teeth. Can the authors discuss this more in-depth? I noticed that the author used Fe²⁺ instead of Fe³⁺ in the precursor, why?

We only added iron (II) sulphate to the organotypic cultures of limpet teeth derived from the FZ (Figure 1i, Supplementary video 3) to see if the teeth would take up the iron from the medium. Iron (II) sulphate was used as it is easily soluble. However, we did not add any iron to cultures of isolated cells or in experiments where conditioned media was to be collected and used for mineralisation. Therefore, haematite is present in chitin scaffolds incubated with conditioned media taken from limpet cells grown in L15-ASW (without with the addition of any iron to the media). This means that the only iron present would be that already within the cells at the point of harvesting from the limpets themselves. To clarify the absence of iron in the media for these experiments the text has been altered to state that:

“To investigate whether radula cells can mineralise chitin scaffolds in an analogous process, we collected and filter-sterilised conditioned media (CM) from 7-day-old cultures of Stage III and IV radula cells which had been maintained in a basal L15-ASW media with no additional iron.”

For further emphasis we have expanded the existing comment on the formation of haematite to read as follows:

The presence of hematite rather than goethite is intriguing; both goethite and haematite contain ferric iron (Fe³⁺) and it is already established that under certain conditions goethite may transform to haematite^{36,37}.

5. The author claimed they produced mineralized chitin by the conditioned medium of radula in figure 4c-f. However, there is only one mineral particle in the image, can the author provide a larger view of this image?

We agree that the image submitted was insufficient. Therefore, we re-analysed the samples and revised the figure to show clearly biogenic iron crystals in the chitin scaffold (see updated Figure 4 and Supplementary Figure 5).

Minor:

1. P34: "Chiton" should be "chiton"

This was corrected

2. I suggest the authors first provide a limpet image and then a whole radula followed by dissected radula for the readers that are not familiar with the system.

We very much appreciate this suggestion. We have added a supplementary video illustrating the process of radula isolation and clear images of this structure ("Supplementary Video 3") to address this.

3. The micro-CT image is not clearly shown for better indicating the different stages of mineralization. I even suggest moving this image into the main Figures instead of putting into the supplemental materials. Similarly, the micro-CT image in supplemental figure 2 is not clear at all.

The microCT images in Supplemental Figure 2 (now re-named Supplemental Video 3) have been enhanced and the ROI cropped down to the area where the mineralised teeth that have been grown in the lab can be seen more clearly, highlighted within a red circle.

4. What is the difference between figure 1g and figure 1h?

These are both representative images showing lab grown limpet radula and teeth. The only difference is the angle from which the images have been captured, to present these clearly.

5. Which part are the head and tail respectively in figure 1 q-s?

The head is now marked by arrows in 1q. A close up of the head is shown in 1r and of the tail in 1s. The figure legend has been amended accordingly.

6. I suggest to move table 1 into supplemental materials because it is very long and does not contain vital information. Actually, table 3 contains many interesting and important information regarding the formation of the tooth. I suggest the author say more about table 3.

Table 1 has been moved into the supplementary (as 'Supplementary Table 1) and all the remaining supplementary tables have been renumbered accordingly..

We have now provided additional heatmap on differentially expressed genes and commented on this in detail in response to the Major comment (as explained above). this Therefore, this Table (now Supplementary Table 4) is now complementing this additional figure and description of transcriptomic findings.

7. P192-194: "The lack of chitinase activity in FZ, concomitant with high expression of 193 different chitinase genes there may suggest a lack of translation, or the generation of 194 protein precursors requiring activation". In the reference I mentioned by Chuang et al, they used proteomics rather than only transcriptome. In their results, they did not find any chitinase in the mature tooth neither. This may reflect that chitinase is only needed between stage II to III. And actually chitin-binding proteins may play important roles in the formation of the tooth.

We have discussed our findings in the context of this publication by Chuang et al in the revised Discussion section, which now reads as follows:

Of the three chitinase 1 (CHIT1) transcripts, two were highly expressed in FZ with expression decreasing along the radula stages with the third transcript (DN209778) expressed at high levels all along the radula, except for Stage I. The high expression levels of chitinases concomitant with several chitin synthases in FZ may represent a high level of plasticity and remodelling of chitin at this stage. The concomitant increase in chitin synthase and decrease in chitinase 1 expressions in Stage I ought to correspond to a net increase in chitin production.

Interestingly, CHIT1 transcript DN209778 was also expressed in mineral-containing Stages II-IV, suggesting continued chitin remodelling, possibly to accommodate goethite crystal intercalation.

8. About the qPCR results, is the y-axis the relative fold compared to that in the foot tissue?

Yes, it is delta-delta between the housekeeping gene and the foot tissue.

Reviewer #2 (Remarks to the Author):

This manuscript is devoted to the molecular basis of the formation and biomineralization of radular teeth in true limpet *Patella vulgata*. Despite vast data of gastropod radula morphology and functioning, there are no study of a molecular basis of the radula biomineralization. This very professional study was done at a high level and combined various methods entirely adequate for the tasks. New data about genes expression in different radula zones involved in the synthesis and mineralization of the radula were obtained. Mapping of the radular zones and the ratio expression of key genes is a valuated contribution to understanding the formation of the chitinous matrix and the processes of biomineralization.

A very significant result of this study is that the authors cultured the cells of the radular sac and, for the first time, synthesized individual chitinous teeth outside of the organism and biomineralized chitinous scaffold in vitro. These new technologies could give a stimulus to a whole direction in the production of biomimetic materials.

We are very pleased with this positive evaluation and thank the Reviewer for the helpful suggestions on the biological aspects of our work.

I have a few comments:

1. St 48. Since there are no clear definitions of the terms "stages" and "Formation Zone" many questions arise, for example, what is the difference between the formation zone and stage I? If teeth are formed in the formation zone, what happens in stage I? What happens with teeth in the maturation zone (stage II)? Why do you use the term zone for the formation zone and stage for the subsequent zones? It would be nice to imagine the processes that occur in different stages / zones to explain the difference in gene expression. Understanding the processes occurring in different stages / zones can be the basis for explaining the difference in expression. In material and methods (st 474), you refer to Mann 1986: do you identify stages based on his data on tooth mineralization?

Would you please add a full description of the stages?

We tried to adhere to the nomenclature provided in previous studies, albeit some of the descriptions are equivocal. We have restructured the relevant parts of the manuscript to improve the clarity of the descriptions of these structures and included information on the gene expression and their possible relevance for the processes occurring in different parts of the radula, yet trying to avoid speculations. The 'Stages' of the radula were previously defined in Mann, S., Perry, C. C. & Webb, J. Structure, morphology, composition and organization of biogenic minerals in limpet teeth. *Proc. R. Soc. London - Biol. Sci.* (1986) doi:10.1098/rspb.1986.0018. We have revised the text so it now as follows:

"The four developmental stages of teeth²⁰ correspond to immature (Stage I, first 15-20 rows without any signs of mineralisation), early maturing (Stage II, the next 12 rows), late maturing (Stage III, the next 30 rows) and maturing (Stage IV, the remaining rows) (Fig. 1b-e)."

One of the regions that was not named unequivocally at the time we started this project, or perhaps we were not aware of the correct terminology for, was the bulbous sack of soft tissue at the proximal end of the radula. More recently, a manuscript describing this region as the formation zone was published: Vortsepneva, E. & Tzetlin, a. General morphology and ultrastructure of the radula of *Testudinalia testudinalis* (O. F. Müller, 1776) (Patellogastropoda, Gastropoda). *J. Morphol.* 280, 1714–1733 (2019). Therefore, we now adopted this terminology in the revised manuscript. We use the terms Stage for Stages I to IV, as defined by Mann et al. and as this terminology is widely used by all authors.

We are not suggesting that teeth are formed in the Formation Zone. We are instead demonstrating that the FZ gives rise to S1 radula, which contains teeth precursors. We have also shown that cells derived from the FZ can differentiate to develop teeth *in vitro*. Teeth at Stage II begin to acquire mineral, as defined by Mann et al 1986.

1. I didn't find supplementary figures 1-2; you probably mean video 1-2?

Yes. We have corrected the names accordingly.

2. St. 72. Since a whole radular sac was taken for the cell culture, all cells that synthesize the teeth and radular membrane, including odontoblasts, membranoblasts and surrounding epithelia, were transferred to this culture. Thus, the normal tissue composition and tissue interactions, which allow radula to be synthesized, are preserved. In this case, tooth formation in the culture testify about normal functioning of the radular sac interactions more than about the pluripotency. If a structure not characteristic of the radula were formed in this cell culture, then it would be pluripotency. The fact that these cells retain the ability to synthesize a tooth in culture, outside of such a complexly organized structure as the radular

sac, is, in my opinion, much more surprising than pluripotency. I suggest adding a paragraph about this.

Tooth formation occurred with the isolated FZ as an organotypic culture (Fig 1 f-h), or with cells isolated from the FZ alone (Fig 1 t). Cultures combining cells from the FZ and SI-SIV radula formed shapes in culture with head and tail formations, but these did not include teeth (Fig 1q-s). This indicates that a structure (the tooth) not characteristic of the FZ is formed, and as such it suggests pluripotency. Furthermore, further transcriptomic analyses of the FZ identified genes associated with stemness. Therefore, we have expanded this part of the Discussion to include:

The Hes Family BHLH Transcription Factor 1 (*Hes1*) has been associated with stem cells across mammals and invertebrates^{31,32}, which agrees with its expression in FZ, where cells isolated from this region showed clear stem cell potential.

In addition to the previous text:

The extremely powerful regeneration and organogenesis potential of the FZ suggests that it is central to directing the activity and organisation of surrounding cells. This effect is comparable to previous studies in sponges, whereby completely separated cells can re-aggregate to form a new sponge structure³⁵. However, sponges are basal metazoans and lack the complex arrangement of tissues and organs found in gastropods, which highlights the exceptional nature of this self-assembly of the limpet radula cells.

3. St 98. What do you mean when you write radula? All radular sack, including Formation Zone? Please specify.

By radula we mean stages I to IV of the radula as defined by Mann et al 1986, not including FZ, which we consistently refer to as such. We have discussed the nomenclature obstacles above.

4. Both sup figure 5-6 named in the site as fig 4.

We apologize for this error, which has been corrected in the revised manuscript.

5. It is important to add a total number of studied specimens to material and methods.

The text has been changed to emphasise the 3R and ARRIVA principles that were applied to our research, wherever possible and it reads as follows:

“Common limpets (*Patella vulgata*) were harvested locally (XXXXXX) under the University Animal Welfare and Ethical Review Body (AWERB) approval (XXXX). Individual limpets were treated as biological replicates and, where possible, multiple analyses were undertaken in each individual to minimise the number.”

6. Decipher all abbreviations across the manuscript (for example, BCA, st. 569).

BCA has been defined in the text along with all other abbreviations. We are sorry for this basic error.

REVIEWER COMMENTS

Reviewer #1 (Remarks to the Author):

The authors have made substantial changes to improve the quality of the paper. In the revised parts, I only have some minor comments which need to be addressed:

1. "Secreted protein, acidic, rich in cysteine (sprc)" should be "SPARC"
2. "The superoxide radical may affect iron solubility and bio-availability", what do you mean by iron solubility and bioavailability?
3. "DIAC and ARX expressions were specifically enriched in FZ only. SPRC, SODC, GBX2 and FRIS were found from S1 to SIV radula, while ETS1 appeared up-regulated at SIII." This new added sentences had many abbreviations which should be defined and described further.
4. "We identified and compared gene orthologs (Supplementary Figure 2c)". I did not find any comparison in Figure S2c.

Reviewer #2 (Remarks to the Author):

General comments

General remarks relate to two points: the first concerns the authors' incomprehension of the histological processes that take place in the radular sac. The second is an incorrect using of terms regarding pluripotency.

I believe that the elimination of these inaccuracies will lead to accept the manuscript for publication. At this moment I recommend a major revision.

Authors: "We are not suggesting that teeth are formed in the Formation Zone. We are instead demonstrating that the FZ gives rise to S1 radula, which contains teeth precursors. We have also shown that cells derived from the FZ can differentiate to develop teeth in vitro. Teeth at Stage II begin to acquire mineral, as defined by Mann et al 1986".

Mann did not consider or discuss the formation of the tooth at all. All radular stages (I-IV) related to already synthesized teeth. The authors identified the formation zone which is oval shaped in all patellogastropods, but they do not recognize the generally accepted fact that the tooth is formed in this zone.

All numerous studies on the radula formation support that the tooth is synthesized by odontoblasts in the formation zone. Teeth formation is well studied and shown for all groups of gastropods, here are just some references:

Kerth, K. (1979). Electron microscopic studies on radular tooth formation in the snails *Helix pomatia* L.

and *Limax flavus* L. (Pulmonata, Stylommatophora). *Cell and Tissue Research*, 203(2), 283–289.

Kerth, K. (1983). Radulaapparat und Radulabildung der Mollusken. II: Zahnbildung, Abbau und Radulawachstum *Zoologische Jahrbücher. Abteilung*

Mackenstedt, U., & Märkel, K. (1987). Experimental and comparative morphology of radula renewal in pulmonates (Mollusca, Gas-tropoda). *Zoomorphology*, 107(4), 209–239. <https://doi.org/10.1007/BF00312262>

Mackenstedt, U., & Märkel, K. (2001). Radular structure and function. In G. M. Barker (Ed.), *The Biology of Terrestrial Molluscs* (pp. 213–236). CABI Publishing. <https://doi.org/10.1079/9780851993188.0213>

Mikhlina, A., Tzetlin, A., & Vortsepneva, E. (2018). Renewal mechanisms of buccal armature in *Flabellina verrucosa* (Nudibranchia: Aeolidida: Flabellinidae). *Zoomorphology*, 137(1), 31–50. <https://doi.org/10.1007/s00435-017-0370-y>

Mischor, B., & Märkel, K. (1984). Histology and regeneration of the radula of *Pomacea bridgesi* (Gastropoda, Prosobranchia). *Zoomorphology*, 104, 42–66

Peters, W. (1979). Basal bodies in the odontoblasts of the limpet, *Patella coerulea* L. (Gastropoda). *Cell and Tissue Research*, 202, 295–301.

Odontoblasts are usually highly specialized cells that are located at the blind end of the radular sac. One group (or one cell) of odontoblasts forms one tooth. Odontoblasts determine the shape of the tooth as well. Membranoblasts are cells located near the odontoblasts, which also have a well-developed synthetic apparatus. Membranoblasts synthesize radular membrane. In addition to these cells, in the lateral parts of the formation zone located a pool of undifferentiated cells with high mitotic activity. Most probably these cells are responsible for replenishing the cellular composition in the formation zone. The incomprehension of the histological processes by the authors leads to errors in the interpretation of the tooth formation. Probably due to these dividing undifferentiated cells that remained in the FZ culture, a new tooth was able to form.

However, from the text of the ms, I still did not understand how regular teeth are synthesized in the culture? is it possible to repeat the results of this experiment with the formation of a tooth?

RESULTS

St 67

What is the teeth precursor? I meet this term for the first time. Please add the definition.

St 77

Authors: “ When the isolated FZ is maintained in vitro, it undergoes regeneration, producing a new ribbon of Stage I radula (Fig. 1f-h)”.

How long did it take for the radula to grow? How many new rows appeared during the observation period? Since one of the versions of the movement of the radula from the formation zone to the working area is the migration of the subradular epithelium, can it simply be the migration of already formed teeth along with the subradular epithelium? Give additional criteria: how many rows of radula formed during the time of observations.

St 90

Pluripotency is a very strict biological term, for example, «Pluripotent cells, such as embryonic blastomeres, differentiate into mature cell types spanning three germ layers (1–3). Although essential for development, pluripotent cells are generally not known to be present in adult animals (4, 5). Adult tissues, by contrast, are typically maintained by specialized, tissue-specific adult stem cells (5–11)».

Wagner, D. E., Wang, I. E., & Reddien, P. W. (2011). Clonogenic neoblasts are pluripotent adult stem cells that underlie planarian regeneration. *Science*, 332(6031), 811–816.

<https://doi.org/10.1126/science.1203983>

According to the criteria for definition in the case described in the article, this is not pluripotency. I strongly recommend removing pluripotency as it is not biologically correct. I propose to choose a more correct term: tissue-specific multipotency or transdifferentiation.

As I wrote above, in any radular sac there are undifferentiated cells with mitotic activity that could produce a new odonto- and membranoblasts as well as for sub- and supradular epithelia. These cells are best candidates for tissue-specific adult stem cells demonstrating a multipotency.

DISCUSSION

St 266

“after two weeks assembled into structures mirroring the shape of the entire radula”

I didn't find the structures mirroring the shape of the entire radula in the results. Could you please describe this in more details?

St 263-270

In the same paragraph authors write: «demonstrated a remarkable stem cell potential», and then: « It remains to be established whether this process involved migration and aggregation of the isolated cells or proliferation of specific cells and their arrangement in response to specific cues».

It is more honest, of course, to say that at this stage of research it is not clear for sure whether this is self-assembly or differentiation potential of specific cells. In addition, there are candidates for these tissue specific stems cells in the formation zone (I've wrote above).

MATERIAL AND METHODS

I still don't understand how many radular sacs were taken for the culture and how many replications of the experiments were done by authors. How many times did you observe synthesis a new tooth in the culture? Is it a reproducible phenomenon? Please add the details to materials and methods and to the results. In case of irreproducibility, there are doubts whether it was a contamination? Could it be that an already formed tooth accidentally got along with the cells?

St 381. How many formations zoned authors used?

The same for st 385. And St 470

REFERENCES

Please check the references carefully. For example, the ref 16 is incorrect.

1. St 635

Memoirs: The Molluscan Radula: its Chemical Composition, and Some Points in its Development. *J. Cell Sci.* (1907)

Should be

Sollas, I. B. J. (1907). The Molluscan Radula: its Chemical Composition, and Some Points in its Development. *Journal of Cell Science*, s2-51(201), 115–136. <https://doi.org/10.1242/jcs.s2-51.201.115>

2. St 609

HA, L. Goethite in Radular Teeth of Recent Marine Gastropods. *Science* 137, 279–280 (1962).

Should be

Lowenstam, H. A. (1962). Goethite in radular teeth of recent marine gastropods. *Science, New Series*, 137(3526), 279–280. <http://rspb.royalsocietypublishing.org/cgi/doi/10.1098/rspb.1989.0052>

Manuscript NCOMMS-21-37434-T

"Biomimetic generation of the strongest known biomaterial found in limpet tooth"

REPLY TO REVIEWER COMMENTS

We would like to thank both Reviewers for their additional constructive and insightful suggestions, which helped us to improve our manuscript.

REVIEWER COMMENTS

Reviewer #1 (Remarks to the Author):

The authors have made substantial changes to improve the quality of the paper. In the revised parts, I only have some minor comments which need to be addressed:

1. "Secreted protein, acidic, rich in cysteine (sprc)" should be "SPARC"

We are grateful for noticing this error, which has now been corrected.

2. "The superoxide radical may affect iron solubility and bio-availability", what do you mean by iron solubility and bioavailability?

We have provided further clarification in the revised manuscript. This paragraph now reads:

In natural aquatic environments the superoxide anion acts as a reductant, while the radical acts as an oxidant and, as such, it has the potential to switch iron between water soluble Fe^{2+} and non-soluble Fe^{3+} , and the latter may find its way into cells³⁴. Thus, the superoxide radical

may affect iron solubility and bio-availability³⁹, and the presence of *sodC*, which functions to remove superoxides, may be important for stable iron deposition within the limpet teeth

3. “DIAC and ARX expressions were specifically enriched in FZ only. SPRC, SODC, GBX2 and FRIS were found from SI to SIV radula, while ETS1 appeared up-regulated at SIII.” This new added sentences had many abbreviations which should be defined and described further.

These missing names have now been added and the revised paragraph reads:

Expression of the morphogenic patterning gene Aristaless Related Homeobox (*ARX*) and the chitin binding protein Di-N-acetylchitobiase (*DIAC*) were specifically enriched in FZ only. An additional morphogenic patterning gene, Gastrulation Brain Homeobox 2 (*GBX2*) was found from SI to SIV radula, together with genes encoding the iron storage protein Ferritin (*FRIS*), the enzyme Superoxide dismutase [Cu-Zn] (*SODC*) and the extracellular matrix protein, acidic and rich in cysteine (*SPARC*). The transcription factor Erythroblast Transformation Specific 1 (*ETS1*) transcription appeared up-regulated at SIII. There are two subsets of transcript encoding chitin-binding PIF proteins; one subset (*PIF_1*) is more highly expressed in SIV radula than elsewhere, while another (*PIF_2*) is predominantly expressed in muscle. *HES1* gene expression was also markedly upregulated at Stage IV.

4. “We identified and compared gene orthologs (Supplementary Figure 1e)”. I did not find any comparison in Figure S2c.

We are grateful for identifying this mistake. It should read: Supp Fig 1e and this has now been corrected in the text.

Reviewer #2 (Remarks to the Author):

General comments

General remarks relate to two points: the first concerns the authors' incomprehension of the histological processes that take place in the radular sac. The second is an incorrect using of terms regarding pluripotency.

I believe that the elimination of these inaccuracies will lead to accept the manuscript for publication. At this moment I recommend a major revision.

Authors: “We are not suggesting that teeth are formed in the Formation Zone. We are instead demonstrating that the FZ gives rise to S1 radula, which contains teeth precursors. We have also shown that cells derived from the FZ can differentiate to develop teeth in vitro. Teeth at Stage II begin to acquire mineral, as defined by Mann et al 1986“. We have Mann did not consider or discuss the formation of the tooth at all.

All radular stages (I-IV) related to already synthesized teeth.

REPLY: These explanations cited above were not present in the manuscript text but in the reply to the previous comments. There, we tried to explain that we were not suggesting that mineralised teeth are formed in the FZ but just new teeth chitin scaffolds or, as we called these, “precursors” emerge with the S1 radula. Therefore, Mann et al, citation was in relation to teeth mineralisation not formation, which occurs in subsequent radula stages. We are sorry for the confusion that we caused. N.B., we prefer “formation” to “synthesis”, when referring to the process of tooth generation. We have also cited the recommended paper by Kerth (Electron microscopic studies on radular tooth formation in the snails *Helix pomatia* L. and *Limax flavus* L. (Pulmonata, Stylommatophora). *Cell Tissue Res.* 203, 283–289 (1979)), before listing the mineralisation stages described by Mann et al.

“The FZ must continuously generate teeth throughout the limpet lifespan, and it has been shown in other gastropods that teeth emerge from this region in their final form²⁰. In the limpet, the four distinct stages of teeth²¹”

The authors identified the formation zone which is oval shaped in all patellogastropods, but they do not recognize the generally accepted fact that the tooth is formed in this zone.

REPLY: We believe, we made no statement in the manuscript indicating that we disagree with the accepted facts. For clarification, we have now included the paper by Kerth 1979 stating that teeth emerge in their final form from this zone. While it is generally established that the tooth is formed in the FZ and subsequently becomes mineralised, our molecular and functional data on chitin production, including the expression and activity of chitin synthases and relative chitin levels, suggest that the chitinous tooth formation and/or modulation may also continue after it leaves the FZ. Therefore, our data identified a new facet to this process and certainly did not aim to contradict previous histological studies.

All numerous studies on the radula formation support that the tooth is synthesized by odontoblasts in the formation zone. Teeth formation is well studied and shown for all groups of gastropods, here are just some references:

REPLY: We have elaborated on the tooth formation, as per specific paragraphs included below, and cited the relevant additional references suggested by the Reviewer 2 here. The only one of these papers we have not cited is Kerth et al 1983, as it is written in German and we have been unable to find a suitable translation.

Kerth, K. (1979). Electron microscopic studies on radular tooth formation in the snails *Helix pomatia* L. and *Limax flavus* L. (Pulmonata, Stylommatophora). *Cell and Tissue Research*, 203(2), 283–289.

Kerth, K. (1983). Radulaapparat und Radulabildung der Mollusken. II: Zahnbildung, Abbau und Radulawachstum *Zoologische Jahrbücher. Abteilung*

Mackenstedt, U., & Märkel, K. (1987). Experimental and comparative morphology of radula renewal in pulmonates (Mollusca, Gas- tropoda). *Zoomorphology*, 107(4), 209–239. <https://doi.org/10.1007/BF00312262>

Mackenstedt, U., & Märkel, K. (2001). Radular structure and function. In G. M. Barker (Ed.), *The Biology of Terrestrial Molluscs* (pp. 213–236). CABI Publishing. <https://doi.org/10.1079/9780851993188.0213>

Mikhlina, A., Tzetlin, A., & Vortsepneva, E. (2018). Renewal mechanisms of buccal armature in *Flabellina verrucosa* (Nudibranchia: Aeolidida: Flabellinidae). *Zoomorphology*, 137(1), 31–50. <https://doi.org/10.1007/s00435-017-0370-y>

Mischer, B., & Märkel, K. (1984). Histology and regeneration of the radula of *Pomacea bridgesi* (Gastropoda, Prosobranchia). *Zoomorphology*, 104, 42–66

Peters, W. (1979). Basal bodies in the odontoblasts of the limpet, *Patella coerulea* L. (Gastropoda). *Cell and Tissue Research*, 202, 295–301.

Odontoblasts are usually highly specialized cells that are located at the blind end of the radular sac. One group (or one cell) of odontoblasts forms one tooth. Odontoblasts determine the shape of the tooth as well.

Membranoblasts are cells located near the odontoblasts, which also have a well-developed synthetic apparatus. Membranoblasts synthesize radular membrane.

In addition to these cells, in the lateral parts of the formation zone located a pool of undifferentiated cells with high mitotic activity. Most probably these cells are responsible for replenishing the cellular composition in the formation zone.

The incomprehension of the histological processes by the authors leads to errors in the interpretation of the tooth formation. Probably due to these dividing undifferentiated cells that remained in the FZ culture, a new tooth was able to form.

REPLY: We reported, for the first time, the fascinating phenomenon that dissociated cells have the capability to form organised structures in culture *in vitro* and even can form teeth in the absence of any supporting organ structures. However, having no anatomical reference points or specific cell markers, we were hesitant to hypothesize on the types of cells in a mixture of enzymatically dissociated cells involved in the tooth formation in culture. In the revised manuscript we have expanded this section of the Discussion to include the description of cells present in FZ and also incorporated the helpful suggestion made by the Reviewer, that undifferentiated cells with high mitotic activity located in the lateral parts of the formation zone are most probably responsible for tooth formation.

However, from the text of the ms, I still did not understand how regular teeth are synthesized in the culture? Is it possible to repeat the results of this experiment with the formation of a tooth?

REPLY: The results of this experiment were reproducible. Ribbons of teeth were grown from organotypic cultures using 36 FZ over 6 independent replicates. Individual teeth have grown in long-term cell cultures of cells dissociated from FZ in three independent replicates (cells maintained for 6 weeks in culture). These individual limpet teeth emerged from what looked like small clusters or even individual cells attached to the culture dish (Please see the additional figure attached for the perusal by the Reviewers). These teeth were quite homogenous in size and shape and resembled those in the radula, but were significantly smaller. These teeth could not be a result of contamination as none were found immediately after cell plating but they emerged after about 6 weeks of cell culture. Furthermore, teeth could not have entered via media changes as ASW and L15-ASW were pre-filtered through Filtropur V50 with 0.1µm pore size and filtered again immediately before use through 0.2µm syringe filters, in order to prevent contamination. The relevant revised text reads:

Results:

Remarkably, a suspension of cells isolated from the whole radula (FZ to Stage IV) grown on haemolymph-coated glass, assembled, after two weeks, into structures with notable 'head' and 'tail' regions (Fig. 1q-s), in a process akin to neo-organogenesis. Furthermore, cultures of cells isolated from FZ and maintained for 6 weeks in culture spontaneously generate individual limpet teeth. These teeth emerged from what looked like small clusters or even individual cells attached to the culture dish, were resembling those formed in the radula but significantly smaller (~20 µm), yet morphologically similar in size and shape (Fig. 1t).

Discussion:

Furthermore, cultures of individual cells isolated from FZ and maintained for 6 weeks in culture spontaneously generated individual limpet teeth. This ability to create an entire limpet tooth in culture, in the absence of the complex multicellular organ such as FZ, demonstrated a remarkable stem cell potential. Of the known types of cells in FZ, membranoblasts generate the radular membrane while odontoblasts are highly specialized

cells that individually or in groups forms one tooth^{19,30}. It is possible that odontoblasts are long lived and were responsible for tooth formation *in vitro*³¹. Another possibility is that undifferentiated, highly mitotic cells found in the lateral part of FZ, presumed to be replenishing the odontoblast and membranoblast populations, are the source of cells responsible for tooth formation *in vitro*²⁴.

Materials and Methods:

For organotypic cultures, 36 isolated Formation Zones (FZ) were studied in 6 independent experiments. Dissected FZ were incubated immediately in 24 well plates containing 1 mL/well of L15-ASW. Organotypic cultures were given 50% media changes on days 2 and 7. Twelve FZ were used in experiments with iron sulphate supplementation, where fresh media containing between 400 nM and 2 μ M Fe(II)SO₄ were applied.

and

All cultures were repeated with at least three independent replicates.

RESULTS

St 67

What is the teeth precursor? I meet this term for the first time. Please add the definition.

REPLY: By a “tooth precursor” we meant nascent, new tooth. We appreciate that this additional term might be confusing and therefore we removed it in the revised manuscript.

St 77

Authors: “When the isolated FZ is maintained *in vitro*, it undergoes regeneration, producing a new ribbon of Stage I radula (Fig. 1f-h)”.

How long did it take for the radula to grow?

How many new rows appeared during the observation period?

REPLY: Rows of teeth emerge from the FZ after 5 to 7 days in culture, and the new radula continues to grow, so that at two weeks there were up to 12 rows of teeth. (Please see the additional figure attached for the perusal by the Reviewers). The revised text reads:

When the isolated FZ is maintained *in vitro*, it starts producing a new ribbon of Stage I radula (Fig. 1f-h). It emerges from FZ after 5 to 7 days in culture, and the new radula continues to grow so that at two weeks there are up to 12 rows of teeth. In the presence of Fe(II)SO₄ in the medium teeth undergo spontaneous mineralisation (Fig. 1i, Supplementary Video. 3).

Since one of the versions of the movement of the radula from the formation zone to the working area is the migration of the subradular epithelium, can it simply be the migration of

already formed teeth along with the subradular epithelium? Give additional criteria: how many rows of radula formed during the time of observations.
St 90

REPLY: This radula generation *ex vivo* is a novel finding. It confirms the unique ability of gastropod tissues to regenerate in culture but does not explain the underlying mechanism. The suggested migration of the subradular epithelium within these FZ in culture might provide some explanation for this phenomenon. However, it is unlikely to be behind the growth of radula with up to 12 rows of teeth, which we observed in our studies. Nevertheless, we included this possibility in the Discussion of the revised manuscript. It reads:

While a few teeth could be simply brought out by migration of the subradular epithelium, the observed generation of a ribbon with 12 rows of teeth makes this explanation unlikely. This radula generation *ex vivo* is a novel finding, which confirms the unique ability of gastropod tissues to regenerate in culture ²⁹.

Pluripotency is a very strict biological term, for example, «Pluripotent cells, such as embryonic blastomeres, differentiate into mature cell types spanning three germ layers (1–3). Although essential for development, pluripotent cells are generally not known to be present in adult animals (4, 5). Adult tissues, by contrast, are typically maintained by specialized, tissue-specific adult stem cells (5–11)». Wagner, D. E., Wang, I. E., & Reddien, P. W. (2011). Clonogenic neoblasts are pluripotent adult stem cells that underlie planarian regeneration. *Science*, 332(6031), 811–816. <https://doi.org/10.1126/science.1203983>
According to the criteria for definition in the case described in the article, this is not pluripotency. I strongly recommend removing pluripotency as it is not biologically correct. I propose to choose a more correct term: tissue-specific multipotency or transdifferentiation. As I wrote above, in any radular sac there are undifferentiated cells with mitotic activity that could produce a new odonto- and membranoblasts as well as for sub- and supradular epithelia. These cells are best candidates for tissue-specific adult stem cells demonstrating a multipotency.

REPLY: As suggested, in the revised manuscript we have removed references to pluripotency. Instead, we have added a paragraph in the Discussion, which evokes the highly mitotic cells present in the FZ. We agree with the Reviewer that these cells are most likely candidates for being tissue-specific adult stem cells with multipotency. This fragment reads:

Another possibility is that undifferentiated, highly mitotic cells found in the lateral part of FZ, presumed to be replenishing the odontoblast and membranoblast populations, are the source of multipotent cells responsible for tooth formation *in vitro* ²⁴.

DISCUSSION

St 266

“after two weeks assembled into structures mirroring the shape of the entire radula”

I didn't find the structures mirroring the shape of the entire radula in the results. Could you please describe this in more details?

REPLY: We were struck by the structural semblance of these de novo structures containing a distinct “head” and an elongated “tail” to the structure of isolated radula. But we agree that it may be purely coincidental. As this resemblance is of no significance to the data presented, we removed this suggestion from the revised version of Discussion. The revised paragraph reads:

Equally remarkably, a suspension of cells isolated from the whole radula, after two weeks assembled into structures containing a distinct bulbous “head” and an elongated “tail”. It is currently not known whether these structures formed in a process involving migration and aggregation of the isolated cells or rather proliferation of specific cells, such as those aforementioned undifferentiated, mitotic FZ cells, and their subsequent precise arrangement in response to some specific cues.

St 263-270

In the same paragraph authors write: «demonstrated a remarkable stem cell potential», and then: « It remains to be established whether this process involved migration and aggregation of the isolated cells or proliferation of specific cells and their arrangement in response to specific cues».

It is more honest, of course, to say that at this stage of research it is not clear for sure whether this is self-assembly or differentiation potential of specific cells. In addition, there are candidates for these tissue specific stems cells in the formation zone (I've wrote above).

REPLY: We have re-phrased this fragment in the Discussion to clarify that we currently do not know the mechanism behind the formation of these highly structured cellular arrangements formed by dissociated d cells in vitro. The revised text reads:

It is currently not known whether these structures formed in a process involving migration and aggregation of the isolated cells or rather proliferation of specific cells, such as those aforementioned undifferentiated, mitotic FZ cells, and their subsequent precise arrangement in response to some specific cues.

MATERIAL AND METHODS

I still don't understand how many radular sacs were taken for the culture and how many replications of the experiments were done by authors.

For organotypic cultures, 36 isolated Formation Zones (FZ) were studied in 6 independent experiments. Dissected FZ were incubated immediately in 24 well plates containing 1 mL/well of L15-ASW. Organotypic cultures were given 50% media changes on days 2 and 7. Twelve FZ

were used in experiments with iron sulphate supplementation, where fresh media containing between 400 nM and 2 μ M Fe(II)SO₄ were applied.

How many times did you observe synthesis a new tooth in the culture?

Is it a reproducible phenomenon?

Please add the details to materials and methods and to the results.

In case of irreproducibility, there are doubts whether it was a contamination?

Could it be that an already formed tooth accidentally got along with the cells?
St 381. How many formations zoned authors used?

The same for st 385. And St 470

REPLY: As explained when addressing this general comment above, the results of this experiment were reproduceable. Teeth that formed in cells in culture were identified in three independently isolated sets of FZ cells maintained for 6 weeks in culture. These were individual, perfectly shaped but smaller in size than limpet teeth that are generated in vivo. Please see the additional figure attached for the perusal by the Reviewers. This information has been added to the revised manuscript as follows:

For organotypic cultures, 36 isolated Formation Zones (FZ) were studied in 6 independent experiments. Dissected FZ were incubated immediately in 24 well plates containing 1 mL/well of L15-ASW. Organotypic cultures were given 50% media changes on days 2 and 7. Twelve FZ were used in experiments with iron sulphate supplementation, where fresh media containing between 400 nM and 2 μ M Fe(II)SO₄ were applied.

For cell isolation, whole radulae or individual sections of the radula (Formation Zone and stages I to IV) were mechanically dissociated before 18-hour incubation in Dispase II (10 U/mL) in L15-ASW on a shaker at room temperature. After this incubation step, DNase I (1 U/mL) (Thermo-Fisher) was added to prevent clumping prior to trituration of suspensions. Dispase II and DNase I were inhibited with 1 mM EDTA and suspensions were filtered through 50 μ m sterile cell strainers before centrifugation at 400 g for 5 minutes and resuspension in 1 mL L15-ASW. Cells were counted on C-Chip haemocytometers (LabTech, East Sussex, UK) prior to seeding on either glass coverslips, or tissue culture plastics.

These teeth could not be a result of contamination because of filtering and none were found immediately after cell plating but they emerged after about 6 weeks of cell culture and were clearly attached to living cells. The relevant revised text reads:

Furthermore, cultures of individual cells isolated from FZ and maintained for 6 weeks in culture spontaneously generated individual limpet teeth. This ability to create an entire limpet tooth in culture, in the absence of the complex multicellular organ such as FZ, demonstrated a remarkable stem cell potential. Of the known types of cells in FZ, membranoblasts generate the radular membrane while odontoblasts are highly specialized cells that individually or in groups forms one tooth^{19,30}. It is possible that odontoblasts are long lived and were responsible for tooth formation *in vitro*³¹. Another possibility is that undifferentiated, highly mitotic cells found in the lateral part of FZ, presumed to be replenishing the odontoblast and membranoblast populations, are the source of multipotent cells responsible for tooth formation *in vitro*²⁴.

REFERENCES

Please check the references carefully. For example, the ref 16 is incorrect.

1. St 635

Memoirs: The Molluscan Radula: its Chemical Composition, and Some Points in its Development. J. Cell Sci. (1907)

Should be

Sollas, I. B. J. (1907). The Molluscan Radula: its Chemical Composition, and Some Points in its Development. Journal of Cell Science, s2-51(201), 115–136. <https://doi.org/10.1242/jcs.s2-51.201.115>

2. St 609

HA, L. Goethite in Radular Teeth of Recent Marine Gastropods. Science 137, 279–280 (1962).

Should be

Lowenstam, H. A. (1962). Goethite in radular teeth of recent marine gastropods. Science, New Series, 137(3526), 279–

280. <http://rspb.royalsocietypublishing.org/cgi/doi/10.1098/rspb.1989.0052>

Thank you for notifying these errors. These references have now been corrected.

REVIEWERS' COMMENTS

Reviewer #1 (Remarks to the Author):

I believe the revised paper is of high quality and is ready to be published. This great paper is highly interdisciplinary, which involves molecular biology, biochemistry, cell biology, and material science. The results are not only a significant advance in understanding the formation of limpet teeth but may invoke new research in both fundamental and applied areas (e.g. small organoids, stem cells, biomimetic synthesis of iron materials). We still have much to learn from this small and ancient marine animal.

One comment:

Unlike human teeth, the limpet teeth are commonly worn out and replaced frequently (~several rows per day). The cells in FZ, therefore, should have some capability to proliferate and differentiate quickly, which may be corresponded to undifferentiated cells with high mitotic activity.

A small error:

Line 286: "highly mitotic cells fund in the lateral part of FZ" in which "fund" should be "found".

Reviewer #2 (Remarks to the Author):

The authors have corrected the text accordingly my suggestions. I recommend to accept this manuscript for publication.

"Biomimetic generation of the strongest known biomaterial found in limpet tooth"

REPLY TO REVIEWER COMMENTS

We would like to thank both Reviewers for their constructive and insightful suggestions made throughout the editorial process, which helped us to improve our manuscript.

We have now corrected the spelling error and incorporated the comment made by the Reviewer 1. We are very pleased with the very kind concluding remark on our manuscript made by this Reviewer.

The revised text reads:

Another possibility is that undifferentiated, highly mitotic cells **found** in the lateral part of FZ, presumed to be replenishing the odontoblast and membranoblast populations, are the source of multipotent cells responsible for tooth formation *in vitro*²⁵.

Equally remarkably, a suspension of cells isolated from the whole radula, after two weeks assembled into structures containing a distinct bulbous "head" and an elongated "tail". It is currently not known whether these structures formed in a process involving migration and aggregation of the isolated cells or rather proliferation of specific cells, such as those aforementioned undifferentiated mitotic FZ cells, **which should have capability to differentiate quickly**, and their subsequent precise arrangement in response to some specific cues.